# Quantification of microenvironmental metabolites in murine cancers reveals determinants of tumor nutrient availability

**Mark R Sullivan[1], Laura V Danai[1,2], Caroline A Lewis[3], Sze Ham Chan[3], Dan Y Gui[1], Tenzin Kunchok[3], Emily A Dennstedt[1], Matthew G Vander Heiden[1,4]\*, Alexander Muir[1,5]\***

[1]Koch Institute for Integrative Cancer Research, Department of Biology, Massachusetts Institute of Technology, Cambridge, United States; [2]Department of Biochemistry and Molecular Biology, University of Massachusetts, Amherst, United States; [3]Whitehead Institute for Biomedical Research, Massachusetts Institute of Technology, Cambridge, United States; [4]Dana-Farber Cancer Institute, Boston, United States; [5]Ben May Department for Cancer Research, University of Chicago, Chicago, United States

**\*For correspondence:**
mvh@mit.edu (MGVH);
amuir@uchicago.edu (AM)

**Competing interest:** See
page 20

**Reviewing editor:** Ralph
DeBerardinis, UT Southwestern
Medical Center, United States

**Abstract** Cancer cell metabolism is heavily influenced by microenvironmental factors, including nutrient availability. Therefore, knowledge of microenvironmental nutrient levels is essential to understand tumor metabolism. To measure the extracellular nutrient levels available to tumors, we utilized quantitative metabolomics methods to measure the absolute concentrations of >118 metabolites in plasma and tumor interstitial fluid, the extracellular fluid that perfuses tumors. Comparison of nutrient levels in tumor interstitial fluid and plasma revealed that the nutrients available to tumors differ from those present in circulation. Further, by comparing interstitial fluid nutrient levels between autochthonous and transplant models of murine pancreatic and lung adenocarcinoma, we found that tumor type, anatomical location and animal diet affect local nutrient availability. These data provide a comprehensive characterization of the nutrients present in the tumor microenvironment of widely used models of lung and pancreatic cancer and identify factors that influence metabolite levels in tumors.

DOI: https://doi.org/10.7554/eLife.44235.001

## Introduction

Tumors exhibit altered metabolism compared to non-transformed tissues (*DeBerardinis and Chandel, 2016*). For example, animal limbs transformed with oncogenic viruses exhibit increased glucose uptake and lactate secretion relative to unaffected limbs (*Cori and Cori, 1925*). Some aspects of tumor metabolism, including higher rates of glucose fermentation to lactate, are cell-intrinsic features that are retained when cancer cells are isolated from tumors (*Koppenol et al., 2011*). Indeed, numerous studies have delineated how cell-intrinsic factors such as oncogenic lesions or epigenetic state alter cellular metabolism, causing phenotypes such as increased glycolysis (*Nagarajan et al., 2016*).

However, beyond cell-intrinsic alterations, tumors have modified tissue architecture and an altered tissue microenvironment; these cell-extrinsic factors can also impact the metabolism of tumors (*Muir et al., 2018*). For instance, the metabolic utilization of both glucose and the amino acid glutamine differs between cells growing in culture and murine tumor models (*Davidson et al.,*

**eLife digest** In the body, cancer cells can rely on different nutrients than normal cells, and they can use these nutrients in a different way. What cancer cells consume also depends on what is available in their immediate environment. In a tumor, cells grab nutrients from the 'interstitial' fluid that surrounds them, but what is present in this liquid may vary within tumors arising in different locations. Understanding what nutrients are 'on the menu' in specific tumors would help to target diseased cells while sparing healthy ones, but this knowledge has been difficult to obtain.

To investigate this, Sullivan et al. used a technique called mass spectrometry to measure the amounts of 120 nutrients present in the interstitial fluid of mouse pancreas and lung tumors. Different levels of nutrients were found in the two types of tumors, and analyses showed that what was present in the interstitial fluid depended on the type of cancer cells, where the tumor was located, and what the animals ate. This suggests that cancer cells may have different needs because they are limited in what they have access to.

It remains to be seen whether the nutrients levels found in mouse tumors are the same as those in humans. Armed with this knowledge, it may then be possible to feed cancer cells grown in the laboratory with the nutrient menu that they would have access to in the body. This could help identify new cancer treatments.

DOI: https://doi.org/10.7554/eLife.44235.002

*2016*; *Muir et al., 2017*; *Sellers et al., 2015*; *Tardito et al., 2015*). Further, the metabolic enzymes that cancer cells rely upon for proliferation in culture are different than those that the same cells utilize to support growth and survival in tumors (*Alvarez et al., 2017*; *Possemato et al., 2011*; *Yau et al., 2017*). Thus, cancer cell metabolism is influenced by microenvironmental cues.

Numerous microenvironmental factors affect cancer cell metabolism (*Anastasiou, 2017*; *Bi et al., 2018*; *Muir et al., 2018*; *Wolpaw and Dang, 2018*), including the presence of stromal cells (*Lyssiotis and Kimmelman, 2017*; *Morandi et al., 2016*), tumor acidity (*Corbet and Feron, 2017*; *Persi et al., 2018*), extracellular matrix properties (*DelNero et al., 2018*; *Tung et al., 2015*) and tumor nutrient levels (*Muir and Vander Heiden, 2018*). In particular, environmental nutrient availability is an important regulator of cancer cell metabolism (*Cantor et al., 2017*; *Muir et al., 2017*; *Schug et al., 2015*; *Tardito et al., 2015*). Nutrient differences between standard cell culture and animal tumor models can drive substantial changes in cancer cell metabolism that alter the response of cancer cells to metabolically targeted drugs (*Cantor et al., 2017*; *Gui et al., 2016*; *Muir et al., 2017*; *Palm et al., 2015*), such that drugs that inhibit proliferation of cancer cells in culture fail to exhibit efficacy in tumors derived from the same cells (*Biancur et al., 2017*; *Davidson et al., 2016*). Thus, determining the concentrations of nutrients in the tumor microenvironment is important to understand and therapeutically target cancer cell metabolism.

Tumors and tissues are supplied with nutrients through the vasculature, which filters a nutrient rich fluid from the circulation into the interstitial space of a tissue or tumor (*Wiig and Swartz, 2012*). The interstitial fluid (IF) then perfuses through the tissue or tumor, exchanging nutrients and wastes with cells. IF is then drained from the tissue or tumor via capillaries and the lymphatic system. Thus, cells in tissues or tumors are not necessarily directly exposed to the nutrients in circulating plasma, but instead are exposed to IF nutrient levels. For healthy organs, IF nutrient levels may be similar to circulating nutrient levels, as these tissues are well vascularized, allowing rapid metabolic exchange with the plasma. Indeed, the IF glucose concentration in healthy skin is very similar to that of circulating plasma (*Lönnroth et al., 1987*). In contrast to the functional vessels found in normal tissues, tumors commonly have an abnormal vasculature with few vessels transporting blood (*Fukumura et al., 2010*). This may lead to reduced nutrient delivery and waste exchange between tumor cells and the circulation. Thus, tumor interstitial fluid (TIF) is thought to be nutrient depleted compared to either normal tissues or the circulation, and have increased concentrations of metabolic waste products. Indeed glucose levels in the TIF of some tumors are lower than in the circulation, while lactate levels are higher (*Burgess and Sylven, 1962*; *Gullino et al., 1964*; *Ho et al., 2015*). However, despite the importance of nutrient availability in regulating tumor metabolism and drug

sensitivity, TIF nutrients beyond glucose and lactate have not been comprehensively measured, and the factors that determine TIF composition have not been characterized.

We sought to systematically measure absolute nutrient concentrations in plasma and TIF. To do so, we have utilized a quantitative mass spectrometry-based approach using both external standards and stable isotope dilution of a library of carbon labeled metabolites to determine the concentration of >118 nutrients in biological fluids. We applied this technique to measure nutrient levels in plasma and TIF isolated from autochthonous and transplantation models of murine lung (LUAD) and pancreatic adenocarcinomas (PDAC) driven by activation of Kras and deletion of p53. Interestingly, we found that anatomical location and tumor tissue of origin are both major determinants of TIF nutrient composition. Dietary changes are also reflected in TIF nutrient levels, while introduction of a *Keap1* mutation into LUAD cells had a smaller effect on the composition of the metabolic tumor microenvironment. Collectively, these experiments elucidate the microenvironmental constraints placed upon tumor metabolism by TIF nutrient levels and provide insight into the factors that dictate tumor nutrient availability.

## Results

### Isolation of TIF from murine PDAC tumors

We first focused on isolating and analyzing TIF nutrient levels in pancreatic ductal adenocarcinoma (PDAC), as this tumor type is known to have inadequate vasculature leading to tumor hypoxia (*Koong et al., 2000*) and nutrient deprivation (*Commisso et al., 2013*; *Kamphorst et al., 2015*; *Lyssiotis and Kimmelman, 2017*; *Sherman et al., 2017*; *Sousa et al., 2016*). End-stage tumors were isolated from the *LSL-Kras^{G12D/+}; Trp53^{flox/flox}; Pdx-1-Cre*, (KP-/-C) mouse model of PDAC (*Bardeesy et al., 2006*). This mouse model recapitulates many aspects of the human disease including dense stroma (*Bardeesy et al., 2006*) and alterations in systemic metabolism (*Danai et al., 2018*; *Mayers et al., 2014*). We applied a previously described method to collect TIF; tumors were placed on a fine mesh and subjected to low speed centrifugation (*Figure 1A*) (*Haslene-Hox et al., 2011*). This method has previously been used to study glucose and lactate levels in tumor samples (*Ho et al., 2015*; *Siska et al., 2017*).

Critically, this centrifugation method does not cause lysis of cells when performed on human tumor samples, ensuring that the isolated fluid is interstitial and not intracellular fluid (*Haslene-Hox et al., 2011*). By measuring LDH activity (an intracellular enzyme and marker of intracellular fluid) in KP-/-C tumor, plasma, and TIF samples, we determined that the LDH activity in the entire TIF volume is <1% of LDH activity found in the tumor (*Figure 1B*). Though levels of LDH activity in the PDAC TIF samples were ~20 fold higher than those found in plasma samples, similar or higher LDH activities were recovered in TIF of solid tumors isolated by orthogonal methods (~60–80 fold higher LDH activity in TIF than plasma) (*Burgess and Sylven, 1962*). These data further confirm that TIF isolation by centrifugation does not result in gross cell lysis and that intracellular fluid is not a major component of PDAC TIF.

The TIF isolation method employed requires tumor removal from the animal and centrifugation for 10 min. The time elapsed during this process could alter TIF composition. Faster TIF isolation was not possible, precluding direct assessment of how the time that elapses during TIF isolation impacts metabolites measured. However, to gain insight into whether tumor removal and centrifugation could potentially cause major alterations in tumor metabolism and thus TIF composition, we measured relative intratumoral levels of 112 metabolites in paired PDAC tumor pieces that were dissected from animals and then either frozen immediately or subjected to TIF isolation prior to freezing. Included in this analysis were levels of succinate, xanthine and hypoxanthine, metabolites shown previously to change in response to ischemia (*Chouchani et al., 2014*). Principle component analysis did not lead to separate clustering of immediately frozen or TIF-isolated samples (*Figure 1—figure supplement 1A*). We found no increase in succinate and only modest increases in xanthine and hypoxanthine levels in tumors samples subjected to TIF isolation (*Figure 1—figure supplement 1B–D*), and the changes observed in xanthine and hypoxanthine were much less than those observed in ischemic tissues (*Chouchani et al., 2014*). These data suggest that the TIF isolation method used does not cause substantial changes in intratumoral metabolite levels; however, development

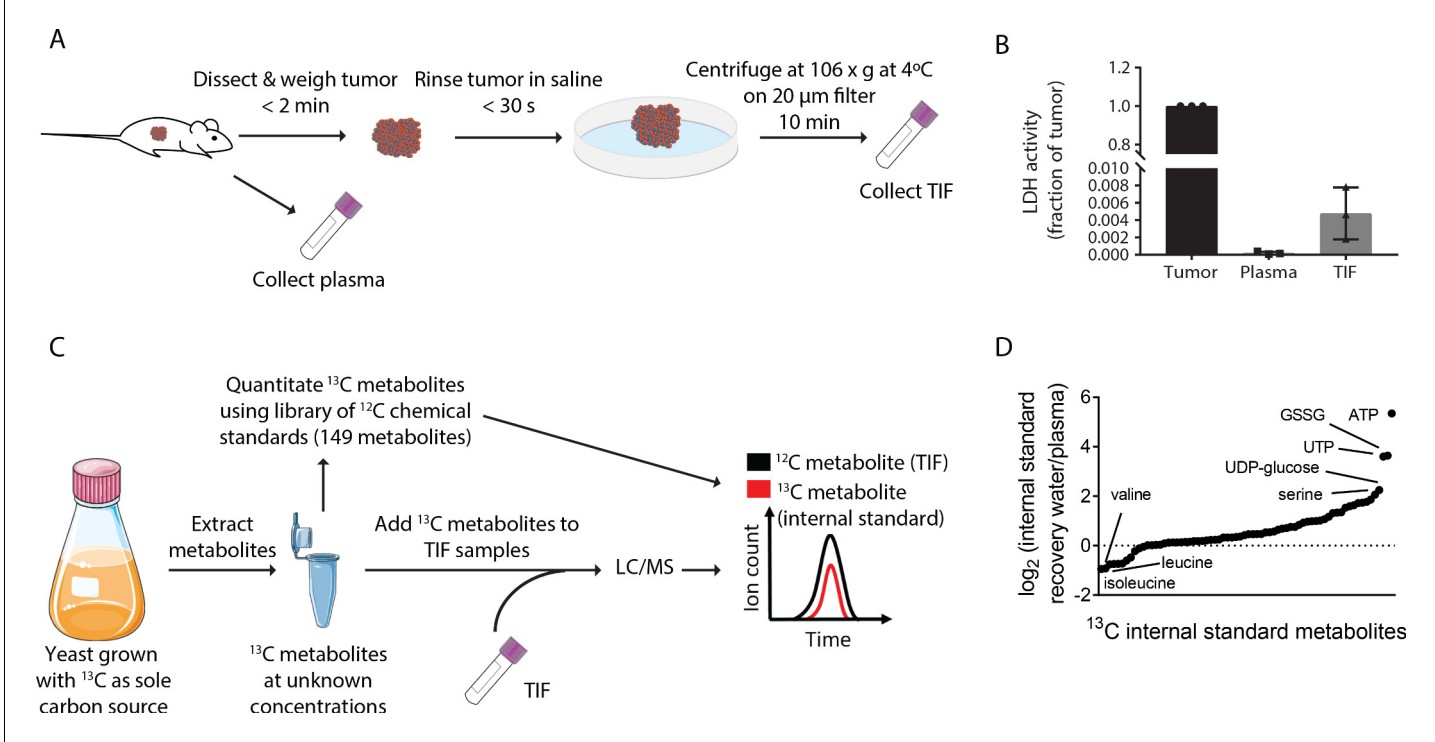

**Figure 1.** Stable Isotope dilution can be utilized to analyze the composition of TIF. (**A**) Schematic of TIF isolation. (**B**) LDH activity assay measuring the amount of LDH present in whole tumors, plasma, and TIF from PDAC tumor bearing mice. LDH activity was calculated for the entire volume of TIF isolated from one tumor and plotted as a ratio to the amount of LDH present in the whole tumor lysate. For each TIF sample, an equal volume of plasma was analyzed and compared to the tumor lysate. n = 3 tumors, plasma, and TIF samples each. (**C**) Schematic summarizing the method used for TIF quantification. (**D**) LC/MS measurement of equal concentrations of 70 $^{13}$C chemical standards suspended in either water or mouse plasma. Data are plotted as the $\log_2$ fold change between the peak area of the metabolite in water versus in plasma. n = 19 plasma samples and n = 6 water samples.
DOI: https://doi.org/10.7554/eLife.44235.003

The following source data and figure supplements are available for figure 1:

**Source data 1.** $^{13}$C metabolite peak areas of metabolites suspended in plasma and water for matrix effect analysis in *Figure 1C*.
DOI: https://doi.org/10.7554/eLife.44235.007

**Figure supplement 1.** TIF isolation causes only minor increases in some metabolites known to be affected by ischemia in tumors.
DOI: https://doi.org/10.7554/eLife.44235.004

**Figure supplement 1—source data 1.** Relative levels of metabolites in paired PDAC tumors that were either dissected and flash frozen, or were frozen after TIF-isolation in *Figure 1—figure supplement 1*.
DOI: https://doi.org/10.7554/eLife.44235.009

**Figure supplement 2.** TIF metabolite quantification is reproducible, particularly when using stable isotope dilution.
DOI: https://doi.org/10.7554/eLife.44235.005

**Figure supplement 2—source data 1.** Metabolite concentrations in 6 PDAC TIF samples analyzed in two inter-day technical replicates in *Figure 1—figure supplement 2*.
DOI: https://doi.org/10.7554/eLife.44235.008

**Figure supplement 3.** Overview of sample preparation and data analysis.
DOI: https://doi.org/10.7554/eLife.44235.006

of faster TIF isolation methods will be required to rigorously evaluate the extent to which the metabolite composition of TIF is altered by the time needed to isolate this fluid.

We attempted to harvest IF from the pancreas, liver and brain of healthy non-tumor bearing animals using the same centrifugation protocol. We were not successful in isolating sufficient IF for analysis, perhaps due to the small size of murine organs and/or the lower interstitial fluid volume and pressure of healthy tissues compared to tumors (*Jain, 1987*). Therefore, our analysis is limited to comparing TIF and plasma from tumor bearing animals.

## The metabolite composition of PDAC TIF differs from plasma

In order to understand the metabolite composition of TIF and plasma samples, we utilized liquid chromatography/mass spectrometry (LC/MS) and both stable isotope dilution and external standard calibration to quantify metabolites (*Figure 1C* and Materials and methods for experimental and data analysis details). We assembled a library of 149 chemical standards of polar metabolites detected in previous studies of human plasma metabolite levels (*Cantor et al., 2017*; *Evans et al., 2009*; *Lawton et al., 2008*; *Mazzone et al., 2016*) (*Supplementary file 1*). We used these standards to quantify $^{13}$C labeled metabolites from yeast extracts derived from cultures where $^{13}$C is the only carbon source. Quantification of these $^{13}$C labeled metabolites allowed them to be used as internal standards for absolute quantification of metabolites by stable isotope dilution (*Figure 1C*). This enabled quantification of 70 metabolites, with the remaining 79 being quantified by external calibration using the library of chemical standards (*Supplementary file 1*).

Absolute LC/MS quantification of metabolites in biological samples such as plasma and TIF is complicated by matrix effects, which alter the ionization and detection of metabolites in different sample types (*Trufelli et al., 2011*). Indeed, when the same amounts of $^{13}$C stable isotope labeled metabolites were added to water or mouse plasma, detected levels of these $^{13}$C standards varied widely depending on whether the metabolites were dissolved in water or plasma (*Figure 1D*). This result confirms the presence of significant matrix effects that confound the comparison of metabolite concentrations measured in different sample types and interfere with absolute quantification by external standard calibration. However, both $^{12}$C and $^{13}$C metabolites are subject to the same sample-dependent ion enhancement or suppression effects, so quantification by stable isotope dilution does not suffer from these potential systematic errors (*Trufelli et al., 2011*). Therefore, we consider the concentrations assigned by external standard calibration to be semi-quantitative, in contrast to the quantitative measurements made by stable isotope dilution. By either isotope dilution or external calibration, we measured between 118–136 metabolites in individual experiments analyzing plasma and TIF with high levels of inter-day reproducibility (stable isotope dilution: Pearson r = 0.9536, p<0.0001; external standard calibration: Pearson r = 0.9516, p<0.0001) (*Figure 1—figure supplement 2*). Low abundance metabolites that were not robustly and reproducibly detected were excluded from individual experiments. Together, this approach allowed us to obtain a combination of quantitative and semi-quantitative measurements of metabolite concentrations in both plasma and TIF.

To determine if the metabolic composition of TIF differs from that of plasma, we isolated TIF and plasma from PDAC tumor bearing mice. Methods of blood collection that require anesthesia can alter circulating metabolite levels (*Overmyer et al., 2015*) (*Figure 2—figure supplement 1*). Therefore, to ensure that plasma samples would be directly comparable to TIF, plasma samples were isolated from mice via cardiac puncture upon euthanasia.

PDAC TIF and plasma samples were profiled using the described metabolomics techniques and grouped by either hierarchical clustering (*Figure 2A*) or principal component analysis (*Figure 2B*) of metabolite concentrations. By each method, the PDAC TIF samples clustered separately from the plasma samples, suggesting that the metabolic composition of PDAC TIF differs from that of plasma. Interestingly, there was substantial heterogeneity in plasma metabolite concentrations between animals. Plasma and TIF was not harvested from mice at a fixed time, and this heterogeneity in plasma metabolites might arise from differences in feed/fast cycle upon harvesting (*Sullivan et al., 2019*) or circadian fluctuations in circulating metabolites (*Abbondante et al., 2016*). PDAC TIF and plasma exhibit different matrix effects (*Figure 2—figure supplement 2*), which could contribute to the observed global differences in metabolite levels between TIF and plasma. Therefore, we also conducted a comparison of PDAC TIF and plasma metabolite levels using only metabolites quantified by stable isotope dilution. PDAC TIF is still substantially different from plasma when analysis is limited to only metabolites quantitated using isotopically labeled internal standards (*Figure 2—figure supplement 6*), suggesting that the global metabolite composition of TIF differs from that of plasma.

The composition of TIF is determined by the summed consumption and excretion rates of nutrients by all cells in the tumor microenvironment, and the exchange rate of those metabolites between the TIF and circulation or lymph; thus, TIF composition does not allow for extrapolation of the rates of consumption and excretion of nutrients from tumors. However, if the exchange rate of

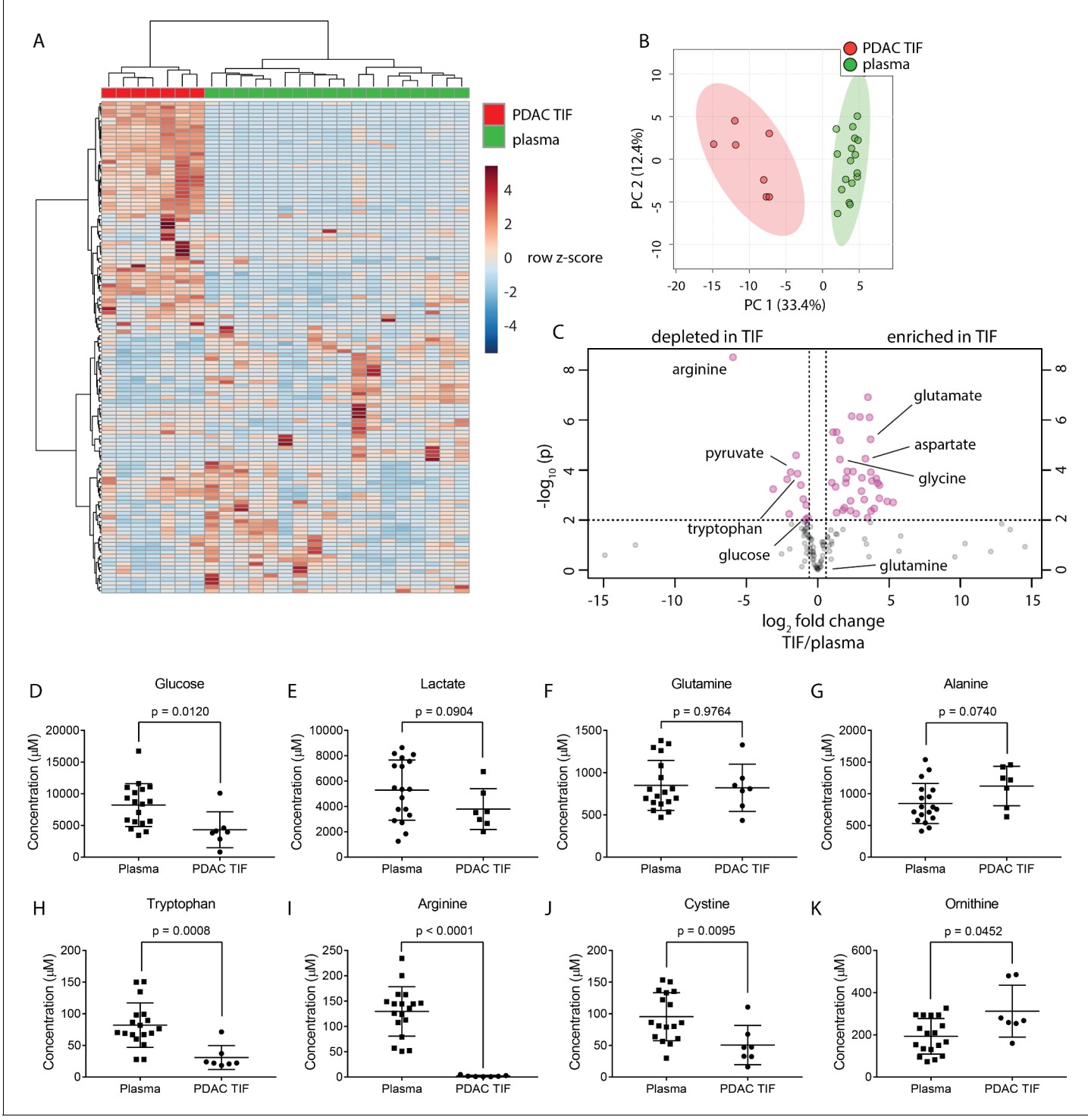

**Figure 2.** TIF metabolite levels are different than those in plasma. (A) Hierarchical clustering of PDAC TIF and mouse plasma samples based on LC/MS measurements of 136 metabolite concentrations. (B) Principal component analysis of PDAC TIF samples and mouse plasma samples based on LC/MS measurements of 136 metabolite concentrations. (C) Volcano plot depicting the $\log_2$ fold change in metabolite concentration between PDAC TIF and plasma. A fold change of 1.5 and a raw p-value of 0.01 assuming unequal variance were used to select significantly altered metabolites indicated in pink. LC/MS measurements of glucose (D), lactate (E), glutamine (F), arginine (G), tryptophan (H), alanine (I), cystine (J), and ornithine (K) in PDAC TIF and plasma samples. p-values were derived from unpaired, two-tailed Welch's t tests for all metabolites except glutamine (F) and tryptophan (H), which did not pass the Shapiro-Wilk test for normal distribution. These p-values were derived from a Wilcoxon-Mann-Whitney test. For all panels, n = 7 for PDAC TIF and n = 18 for plasma samples.

*Figure 2 continued on next page*

*Figure 2 continued*

DOI: https://doi.org/10.7554/eLife.44235.010

The following source data and figure supplements are available for figure 2:

**Source data 1.** Concentrations of 136 metabolites determined by both external standard calibration and stable isotope dilution in all PDAC TIF and plasma samples in *Figure 2*.
DOI: https://doi.org/10.7554/eLife.44235.018

**Figure supplement 1.** Method of blood collection affects plasma metabolite levels.
DOI: https://doi.org/10.7554/eLife.44235.011

**Figure supplement 1—source data 1.** Concentrations of 117 metabolites determined by both external standard calibration and stable isotope dilution for all plasma samples isolated from C57BL/6J mice either by retro-orbital or cardiac puncture collection in *Figure 2—figure supplement 1*.
DOI: https://doi.org/10.7554/eLife.44235.019

**Figure supplement 2.** TIF metabolite levels are different from those in plasma of paired mice.
DOI: https://doi.org/10.7554/eLife.44235.012

**Figure supplement 2—source data 1.** $^{13}$C metabolite peak areas of metabolites suspended in plasma and TIF for matrix effect analysis in *Figure 2—figure supplement 2*.
DOI: https://doi.org/10.7554/eLife.44235.020

**Figure supplement 3.** Loading plot presenting the contribution of individual metabolites to the PCA components in *Figure 2B*.
DOI: https://doi.org/10.7554/eLife.44235.013

**Figure supplement 4.** Hierarchical clustering of PDAC TIF and mouse plasma samples based on LC/MS measurements of 136 metabolite concentrations.
DOI: https://doi.org/10.7554/eLife.44235.014

**Figure supplement 5.** Plasma and TIF exhibit different matrix effects.
DOI: https://doi.org/10.7554/eLife.44235.015

**Figure supplement 6.** PDAC TIF differs from plasma when comparing only those metabolites quantified using internal isotope-labeled standards.
DOI: https://doi.org/10.7554/eLife.44235.016

**Figure supplement 6—source data 1.** Concentrations of 62 metabolites determined only by stable isotope dilution in all PDAC TIF and plasma samples in *Figure 2—figure supplement 6*.
DOI: https://doi.org/10.7554/eLife.44235.021

**Figure supplement 7.** PDAC TIF metabolite levels are not significantly different between large and small tumors.
DOI: https://doi.org/10.7554/eLife.44235.017

TIF and the whole-body circulation is slow or compromised, then nutrients that are highly consumed by cells within a tumor may be depleted in the TIF relative to the circulation, while metabolic by-products may accumulate. Therefore, we predicted that nutrients that are highly consumed by tumors and cancer cells in culture may be depleted in TIF, and metabolites excreted by tumors may accumulate in TIF. Consistent with the avid consumption of glucose observed in tumors and cancer cell lines, glucose was depleted in TIF compared to plasma (*Figure 2C–D*). Amino acids known to be produced by tumors such as glycine and glutamate (*Hosios et al., 2016*; *Jain et al., 2012*) were enriched in TIF (*Figure 2C*). Interestingly, the amino acid glutamine, which is consumed rapidly by cultured cells (*Eagle, 1955*; *Hosios et al., 2016*; *Jain et al., 2012*), was present at similar concentrations in TIF and plasma (*Figure 2C,E*). We also found that alanine, an amino acid reported to support PDAC cells (*Sousa et al., 2016*), was abundantly present in both TIF and plasma (*Figure 2F*). Further, metabolites important for immune cell function such as arginine, tryptophan and cystine (*Geiger et al., 2016*; *Moffett and Namboodiri, 2003*; *Srivastava et al., 2010*) were depleted in PDAC TIF compared to plasma (*Figure 2C,G–I*). Additionally, levels of ornithine in PDAC TIF increased relative to plasma, suggesting that local arginase activity (*Caldwell et al., 2018*) may account for PDAC depletion of arginine (*Figure 2J*). Many metabolites were enriched in TIF (*Figure 2C*), suggesting that the PDAC tumor microenvironment is not depleted for all nutrients. Instead, PDAC TIF is composed of a complex mix of metabolites that are different from those present in circulation.

## Tumor size does not dictate PDAC TIF composition

Having established that PDAC TIF composition is different from that of plasma, we next sought to understand the factors that influence TIF composition. We hypothesized that five factors could influence TIF composition: tumor size, anatomical location, tumor tissue of origin, diet, and tumor

genetics. We used mouse models of PDAC and LUAD to systematically test the impact of these factors on TIF nutrient composition. First, we sought to test if the size of tumors influenced the composition of TIF. Since murine PDAC tumors cause morbidity at different times and tumor sizes, we were able to isolate TIF from end stage PDAC tumors of varying sizes (0.31 g – 2.81 g). We tested if tumor size significantly altered TIF composition by comparing TIF metabolite concentrations between large (1.71–1.24 g) and small (0.78–1.22 g) PDAC tumors (*Figure 2—figure supplement 7*). We found that, at least within this size range of tumors, tumor size does not appear to dictate PDAC TIF metabolite levels.

## Tumor location affects the composition of TIF

We hypothesized that PDAC tumors growing in the pancreas, with its diverse set of stromal cells and poor vascularization, might have different TIF composition than the same tumor cells growing in other organs or anatomical locations in the body. To test this hypothesis, we compared the metabolic composition of TIF from autochthonous KP-/-C PDAC tumors to that of tumors derived by subcutaneously injecting PDAC cells isolated from KP-/-C tumors into genetically identical C57BL/6J mice (*Figure 3A*). We did not observe any relationship between tumor type and the volume of TIF isolated (data not shown). Based on measurement of 123 quantitated metabolites that were detectable and quantifiable in this experiment, PDAC TIF was metabolically distinct from TIF derived from isogenic subcutaneous tumors both by principal component analysis (*Figure 3B*) and by hierarchical clustering (*Figure 3C*). Interestingly, concentrations of tryptophan, arginine and cystine, which are depleted in autochthonous PDAC TIF, were relatively higher in subcutaneous PDAC TIF (*Figure 3D–F*). This suggests that subcutaneous models of PDAC may not mimic the metabolic microenvironment of PDAC tumors. Furthermore, while potentially confounded by comparing transplant and autochthonous PDAC models, these results suggest that the metabolic composition of TIF is not only determined by tumor-intrinsic factors, but potentially also by the anatomical location in which the tumor is growing.

## Dietary changes alter TIF composition

Though TIF metabolite levels do not match those found in plasma, factors that influence circulating nutrient levels may be reflected in TIF. One important determinant of plasma metabolite levels is diet; thus, we examined whether dietary changes would alter TIF metabolite levels. For this analysis, we isolated TIF from isogenic PDAC subcutaneous allografts in mice fed either standard mouse chow derived from plant and animal products or a defined diet that replaces whole protein with purified amino acids (*Supplementary file 2*). This allows us to compare genetically identical tumors growing in the same anatomical location, where only diet is altered. These diets contain many differences in nutrient levels, providing a test case to determine whether significant dietary alterations affect TIF metabolite levels. Based on measurement of 123 metabolites that were detectable and quantifiable in this experiment, TIF from mice fed standard chow differed from TIF from mice fed a defined diet based on principal component analysis (*Figure 4B*) and hierarchical clustering (*Figure 4C*).

We next wanted to determine if the changes in TIF composition between diets were primarily due to altered nutrient availability to TIF from circulation, or if other physiological effects of altered diet were affecting TIF. If dietary perturbations primarily affect TIF by simply altering plasma nutrient levels, then the concentration of a metabolite in TIF should correlate with its concentration in plasma to the same degree in both dietary conditions. Thus, if a dietary change increases the plasma level of a metabolite, then the TIF concentration of that metabolite should also increase. Indeed, there is a strong correlation (Pearson r = 0.8927, p<0.0001) between TIF to plasma ratio of metabolite concentrations between mice on different diets (*Figure 4D*). Furthermore, linear regression of TIF to plasma ratios in mice fed different diets nearly yields an identity function (slope = 0.9511, $R^2$ = 0.7969) (*Figure 4D*). Thus, while individual TIF metabolite levels are scaled by some factor relative to plasma levels, dietary perturbation does not broadly affect this scaling factor. Thus, in addition to local microenvironmental factors, systemic metabolic changes can affect the composition of TIF by altering circulating nutrient levels.

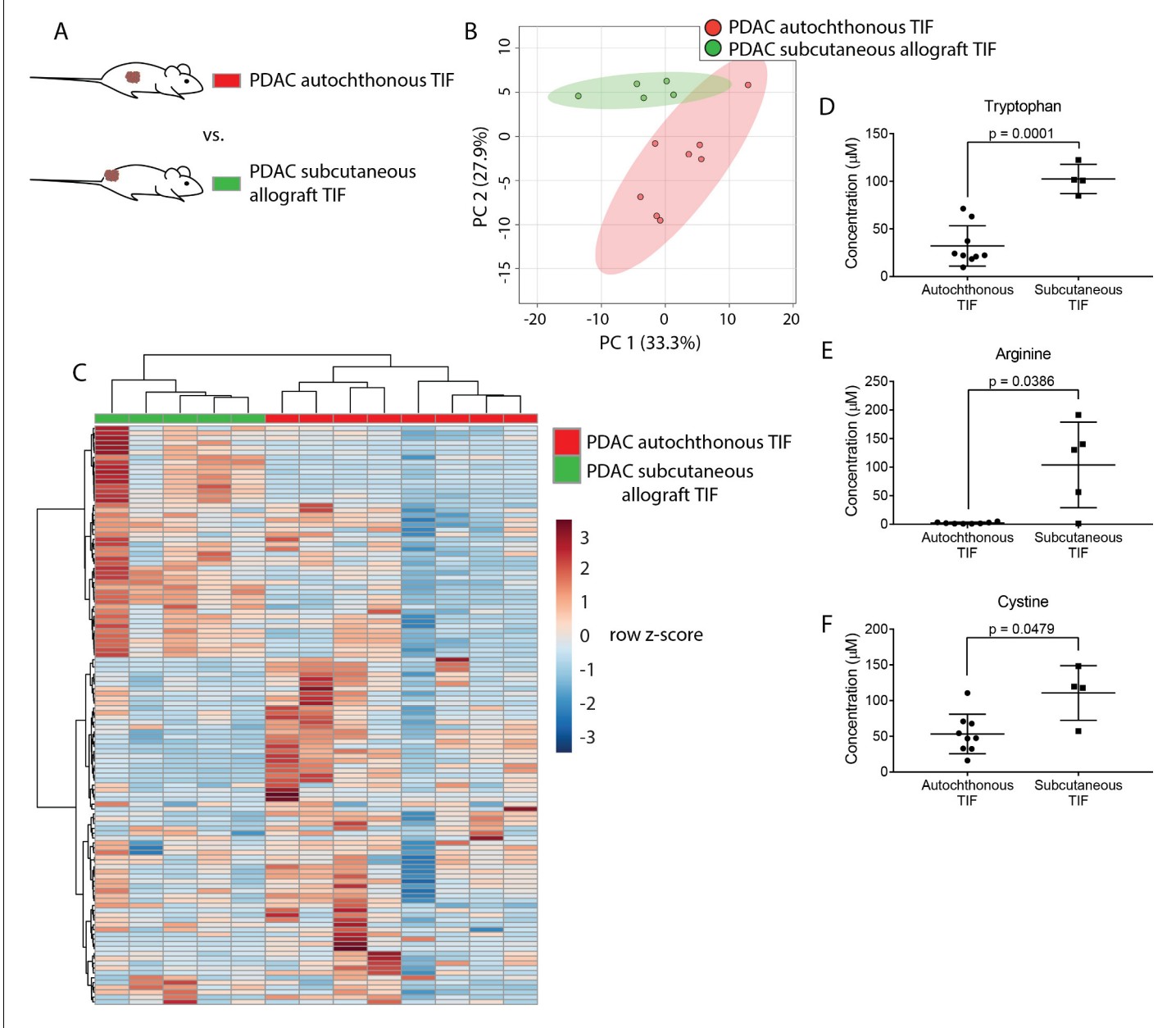

**Figure 3.** Tumor location dictates metabolic TIF composition. (**A**) Diagram of experimental models used to test the effect of tumor location on TIF metabolite levels. Principal component analysis (**B**) and hierarchical clustering (**C**) of PDAC TIF and PDAC subcutaneous allograft TIF samples based on LC/MS measurements of 123 metabolite concentrations. LC/MS measurements of tryptophan (**D**), arginine (**E**), and cystine (**F**) in PDAC TIF and PDAC subcutaneous allograft TIF. p-values were derived from unpaired, two-tailed Welch's t tests. For all panels, n = 7 for PDAC TIF samples and n = 5 for PDAC subcutaneous allografts.

DOI: https://doi.org/10.7554/eLife.44235.022

The following source data and figure supplements are available for figure 3:

**Source data 1.** Concentrations of 123 metabolites determined by both external standard calibration and stable isotope dilution in all autochthonous and subcutaneous PDAC TIF samples in *Figure 3*.

DOI: https://doi.org/10.7554/eLife.44235.025

**Figure supplement 1.** Loading plot presenting the contribution of individual metabolites to the PCA components in *Figure 3B*.

DOI: https://doi.org/10.7554/eLife.44235.023

**Figure supplement 2.** Hierarchical clustering of PDAC TIF and PDAC subcutaneous allograft TIF samples based on LC/MS measurements of 123 metabolite concentrations.

DOI: https://doi.org/10.7554/eLife.44235.024

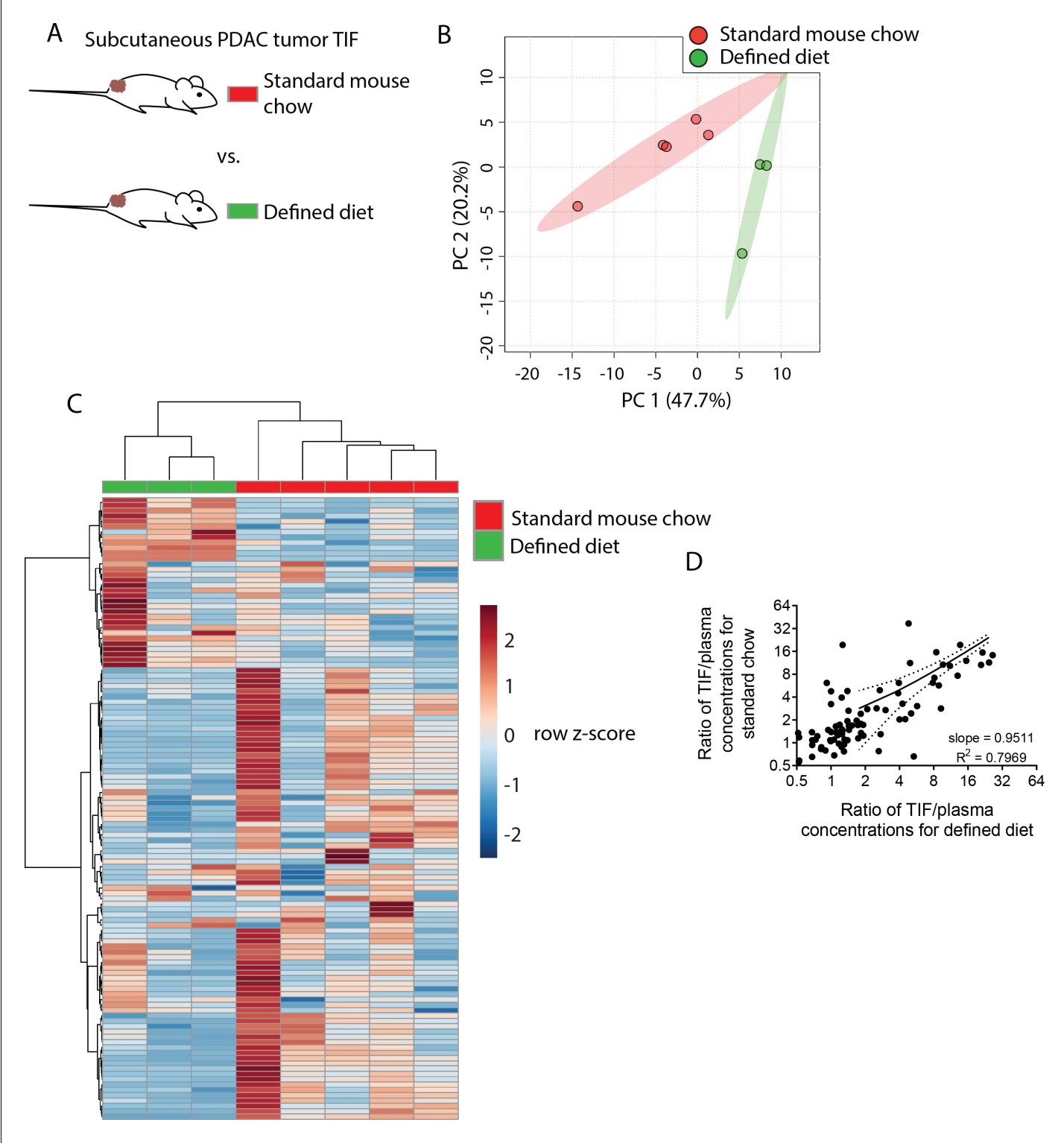

**Figure 4.** Dietary changes alter TIF composition. (**A**) Schematic of experimental models used to test the effect of diet on TIF metabolite levels. Principal component analysis (**B**) and hierarchical clustering (**C**) of subcutaneous PDAC allograft TIF samples from mice fed standard mouse chow versus mice fed a defined diet based on LC/MS measurements of 123 metabolite concentrations. (**D**) Ratios of LC/MS measurements of 123 metabolites in TIF versus matched plasma from the same mouse fed standard mouse chow plotted versus the same ratios in mice fed a defined diet. For all panels, n = 5 for TIF from mice fed standard mouse chow and n = 3 for TIF from mice fed a defined diet.

DOI: https://doi.org/10.7554/eLife.44235.026

*Figure 4 continued on next page*

*Figure 4 continued*

The following source data and figure supplements are available for figure 4:

**Source data 1.** Concentrations of 123 metabolites determined by both external standard calibration and stable isotope dilution in all plasma and subcutaneous PDAC TIF samples in *Figure 4*.

DOI: https://doi.org/10.7554/eLife.44235.029

**Figure supplement 1.** Loading plot presenting the contribution of individual metabolites to the PCA components in *Figure 4B*.

DOI: https://doi.org/10.7554/eLife.44235.027

**Figure supplement 2.** Hierarchical clustering of subcutaneous PDAC allograft TIF samples from mice fed standard mouse chow versus mice fed a defined diet based on LC/MS measurements of 123 metabolite concentrations.

DOI: https://doi.org/10.7554/eLife.44235.028

## Tumor tissue of origin affects TIF makeup

Tumor location and circulating metabolite levels are cell-extrinsic factors that influence the TIF composition. However, there exist many cell-intrinsic factors that alter cancer cell metabolism. For instance, the metabolic properties of cancer cells depend upon the tissue from which they originated (*Hu et al., 2013*; *Mayers et al., 2016*; *Yuneva et al., 2012*). To examine whether tissue of origin influences the metabolic makeup of the tumor microenvironment, cancer cells derived from lung (*Jackson et al., 2005*; *Jackson et al., 2001*) and PDAC (*Bardeesy et al., 2006*) tumors both driven by activation of *Kras* and loss of *Trp53* were injected subcutaneously into C57BL/6J mice, such that tumors were established in the same location and with the same oncogenic driver mutations, but different tissues of origin (*Figure 5A*). Based on measurement of 104 metabolites that were detectable and quantifiable in this experiment, TIF from subcutaneous tumors derived from LUAD clustered separately from PDAC subcutaneous allograft TIF by principal component analysis (*Figure 5B*) and hierarchical clustering (*Figure 5C*). Similar results were obtained when the analysis was limited to only metabolites quantitated by stable isotope dilution (*Figure 5—figure supplement 1*). LUAD and PDAC tumors are known to have different branched-chain amino acid metabolism (*Mayers et al., 2016*), and levels of branched-chain amino acids and their catabolites were different between LUAD and PDAC tumors (*Figure 5D*). Additionally, a number of metabolites involved in thiol metabolism are altered between LUAD and PDAC tumors (*Figure 5D*) (*Gall et al., 2010*; *Irino et al., 2016*), suggesting potential differences in sulfur metabolism between these tumors. Collectively, these data suggest that tissue of origin is a determinant of the metabolic composition of TIF.

## Genetic loss of the tumor suppressor Keap1 does not have a large effect on TIF composition

Genetic alterations can profoundly alter cancer metabolism (*Nagarajan et al., 2016*). As a test case for whether tumor genetics can influence the metabolism of TIF, we focused on the tumor suppressor *Keap1*. *Keap1* loss is a common occurrence in lung cancer that alters expression of oxidative stress response genes and nutrient transporters, which causes cells to secrete high levels of glutamate and renders tumors highly dependent on glutamine catabolism for growth (*Romero et al., 2017*; *Sayin et al., 2017*). Thus, *Keap1* null tumors may possess remodeled metabolism that would be reflected in TIF composition. To test this possibility, we injected previously described LUAD cells with wild-type *Keap1* (sgControl) or *Keap1* loss (sg*Keap1*) (*Romero et al., 2017*) subcutaneously into the flanks of C57BL/6J mice (*Figure 6A*), and TIF was isolated from these tumors. Based on measurement of 131 metabolites that were detectable and quantifiable in this experiment, TIF samples did not cluster separately based on *Keap1* status by principal component analysis (*Figure 6B*), although they could be separated by hierarchical clustering (*Figure 6C*). Surprisingly, anticipated changes in TIF composition based on alterations to cancer cell metabolism by *Keap1* loss, such as decreased glutamine, glucose and cystine, and increased glutamate and lactate (*Romero et al., 2017*) were not observed between TIF samples of *Keap1* wild-type and *Keap1* null tumors (*Figure 6D–H*). Together, these results suggest that genetic *Keap1* status has less of an effect on metabolic TIF content in subcutaneous lung cancer allografts, and that not all cancer cell-intrinsic perturbations of metabolism cause detectable changes to the tumor nutrient milieu.

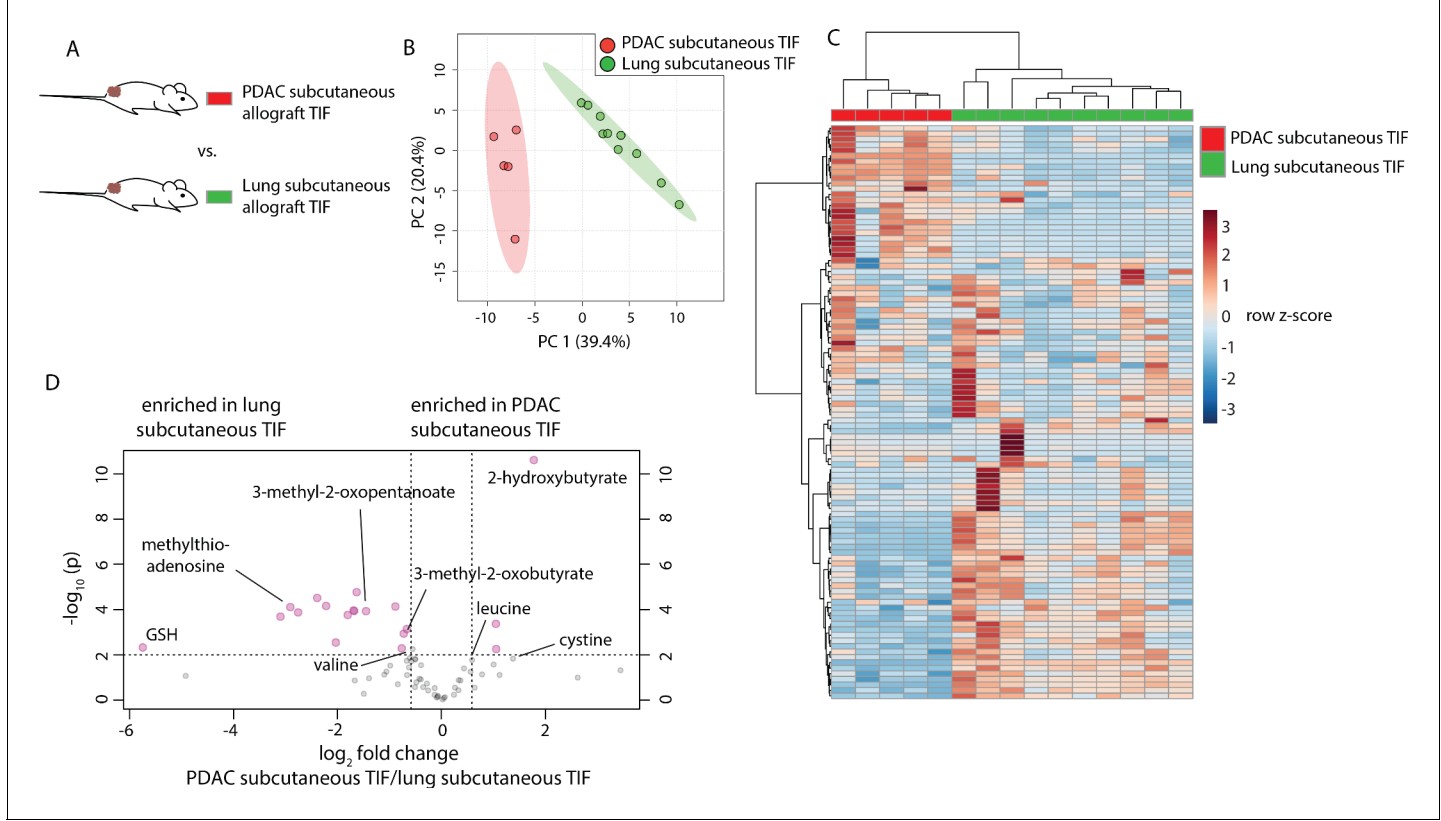

**Figure 5.** Tumor tissue of origin influences TIF composition independent of tumor location. (**A**) Diagram of experimental models used to test the effect of tumor tissue of origin on TIF metabolite levels. Principal component analysis (**B**) and hierarchical clustering (**C**) of PDAC subcutaneous allograft TIF and LUAD subcutaneous allograft TIF samples based on LC/MS measurements of 104 metabolite concentrations. (**D**) Volcano plot depicting the $\log_2$ fold change in metabolite concentration between PDAC and LUAD TIF for metabolites measured using stable isotope dilution. A fold change of 1.5 and raw p-value of 0.01 assuming unequal variance were used to select significantly altered metabolites indicated in pink. For all panels, n = 5 for PDAC subcutaneous allograft TIF samples and n = 10 for LUAD subcutaneous allograft TIF samples.

DOI: https://doi.org/10.7554/eLife.44235.030

The following source data and figure supplements are available for figure 5:

**Source data 1.** Concentrations of 104 metabolites determined by both external standard calibration and stable isotope dilution in all subcutaneous PDAC and LUAD TIF samples in *Figure 5*.

DOI: https://doi.org/10.7554/eLife.44235.034

**Figure supplement 1.** Tumor tissue of origin influences TIF when measured using internal standards only.

DOI: https://doi.org/10.7554/eLife.44235.031

**Figure supplement 1—source data 1.** Concentrations of 66 metabolites determined by only stable isotope dilution in all subcutaneous PDAC and LUAD TIF samples in *Figure 5—figure supplement 1*.

DOI: https://doi.org/10.7554/eLife.44235.035

**Figure supplement 2.** Loading plot presenting the contribution of individual metabolites to the PCA components in *Figure 5B*.

DOI: https://doi.org/10.7554/eLife.44235.032

**Figure supplement 3.** Hierarchical clustering of PDAC subcutaneous allograft TIF and LUAD subcutaneous allograft TIF samples based on LC/MS measurements of 104 metabolite concentrations.

DOI: https://doi.org/10.7554/eLife.44235.033

## Discussion

### Pancreatic and lung tumor cells are exposed to nutrient levels that differ from those found in plasma

Intravital imaging of tumors in patients (*Fisher et al., 2016*) and murine cancer models (*Fukumura et al., 2010*) has revealed that solid tumors generally have abnormal and dysfunctional vasculature and lymphatic vessels. For example, only ~30% of PDAC blood vessels appear functional

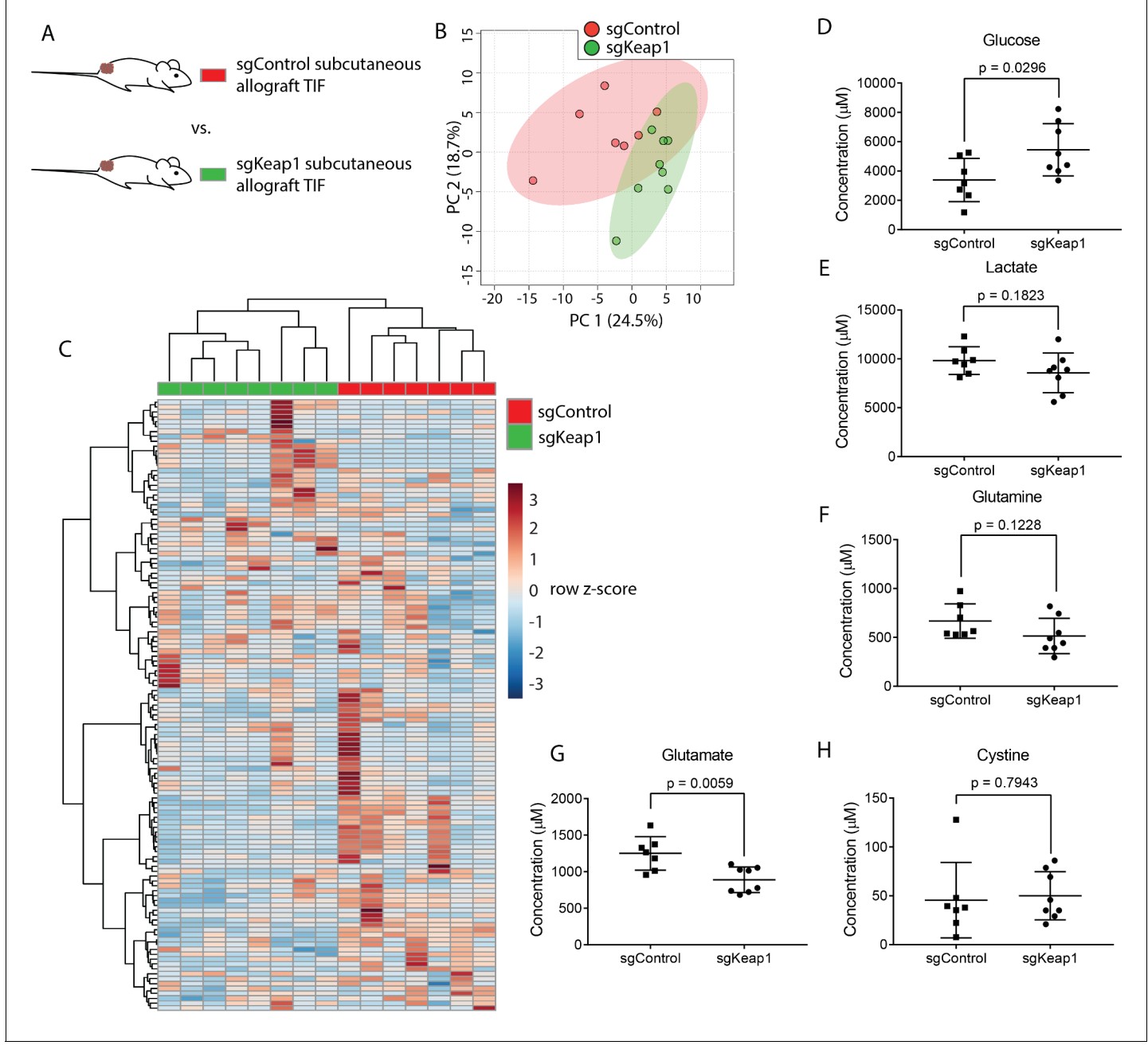

**Figure 6.** Genetic *Keap1* status is not a major determinant of TIF composition in subcutaneous LUAD allograft tumors. (A) Schematic of experimental models used to test the effect of genetic loss of *Keap1* on TIF metabolite levels. Principal component analysis (B) and hierarchical clustering (C) of *Keap1* wild-type (sgControl) and *Keap1* null (sgKeap1) subcutaneous LUAD allograft TIF samples based on LC/MS measurements of 131 metabolite concentrations. LC/MS measurements of glucose (D), lactate (E), glutamine (F), glutamate (G), and cystine (H) in *Keap1* wild-type (sgControl) and *Keap1* null (sgKeap1) subcutaneous lung allograft TIF samples. p-values were derived from unpaired, two-tailed Welch's t tests. For all panels, n = 10 for sgControl subcutaneous lung allograft TIF samples and n = 8 for sgKeap1 subcutaneous LUAD allograft TIF samples.

DOI: https://doi.org/10.7554/eLife.44235.036

The following source data and figure supplements are available for figure 6:

**Source data 1.** Concentrations of 131 metabolites determined by both external standard calibration and stable isotope dilution in all sgControl and sg*Keap1* LUAD TIF samples in *Figure 6*.

DOI: https://doi.org/10.7554/eLife.44235.039

**Figure supplement 1.** Loading plot presenting the contribution of individual metabolites to the PCA components in *Figure 6B*.

DOI: https://doi.org/10.7554/eLife.44235.037

*Figure 6 continued on next page*

*Figure 6 continued*

**Figure supplement 2.** Hierarchical clustering of *Keap1* wild-type (sgControl) and *Keap1* null (sgKeap1) subcutaneous LUAD allograft TIF samples based on LC/MS measurements of 131 metabolite concentrations.

DOI: https://doi.org/10.7554/eLife.44235.038

(*Olive et al., 2009*). This is thought to limit nutrient delivery from the circulation to tumors both by reducing blood flow and by inhibiting transcapillary filtration due to increased interstitial pressure (*Stylianopoulos et al., 2018*). PDAC tumors in particular have striking defects in vasculature with numerous collapsed vessels triggered by increased interstitial pressure (*DuFort et al., 2016*; *Provenzano et al., 2012*) or solid stress (*Chauhan et al., 2014*). In contrast to normal tissues, which, while not in perfect metabolic equilibrium with the circulation (*Cengiz and Tamborlane, 2009*) are reported to have nutrient levels close to what is in the circulation (*Lönnroth et al., 1987*), it is unlikely that the interstitial environment of tumors is in rapid exchange with the circulation. Coupled with the high metabolic rate of cancer cells, this has led to speculation that tumors will be both nutrient deprived and accumulate metabolic wastes. We find that TIF metabolite levels in the murine cancer models assessed are different from the nutrient levels in plasma. Contrary to many assumptions, not all nutrients are depleted in TIF, with levels of some being higher than what is found in circulation. Thus, our measurements of plasma and TIF nutrient composition are consistent with a model where tumors with abnormal vasculature exist largely as a separate compartment from the bulk circulation, and can have different steady state metabolic microenvironments. This underscores the importance of understanding TIF composition for studies of tumor metabolism.

Interestingly, a number of existing and experimental therapies alter the vasculature and interstitial transport properties of tumors to increase tumor perfusion. For example, anti-VEGF therapies are thought to normalize tumor vasculature leading to decreased interstitial pressure and increased perfusion (*Jain, 2001*), and a number of approaches have been developed to limit the stromal compression of tumor blood vessels that increases tumor pressure and decreases tumor perfusion (*Olive et al., 2009*; *Pinter and Jain, 2017*; *Provenzano et al., 2012*). Future studies to manipulate tumor perfusion will determine the extent to which tumor nutrient availability is influenced by stromal desmoplasia and altered tumor vasculature.

While we have considered the tumor as a single compartment and analyzed bulk TIF from the entire tumor, it is possible that the tumor contains a population of sub-compartments with different perfusion rates (*Hensley et al., 2016*) and heterogeneous TIF compositions (*Wagner and Wiig, 2015*). Measurements of intratumor metabolites in subsections of tumors suggests that some tumor regions experience different nutrient levels than others (*Pan et al., 2016*). Thus, while the overall tumor metabolic microenvironment differs from the circulation, an important future challenge will be to understand how compartmentalized nutrient delivery contributes to metabolic heterogeneity observed in solid tumors (*Hensley et al., 2016*).

Though TIF appears to constitute a separate compartment from plasma, tumor nutrients are ultimately delivered from the circulation. Thus, perturbations to systemic nutrient levels by altering diet affect TIF composition. This relationship between plasma and TIF nutrient levels across different diets suggests that TIF levels of metabolites are derived from plasma metabolite concentrations, and changes in plasma metabolite levels lead to scaled changes in TIF metabolite levels. This argues that a major mechanism by which diet influences tumor nutrient availability is by altering nutrients available in the circulation.

## Tumor properties that influence the metabolic microenvironment

Beyond dietary factors, we considered four variables that could influence tumor nutrient availability: tumor size, tissue of origin, tumor anatomical location and tumor genetic make-up. Though we observed that tumor size did not strongly affect the composition of TIF in PDAC tumors, we collected TIF from a relatively small range of sizes of end stage PDAC tumors. Future work to examine the TIF composition of tumors at various sizes and stages of development may reveal effects of tumor size on TIF nutrient levels.

The finding that PDAC and LUAD tumors have different TIF composition suggests that tissue of origin is a factor that can influence the metabolic microenvironment; however, the mechanism(s) driving this difference remain unclear. Whether cancer cell-intrinsic metabolic differences endowed

by the epigenetic memory of the tissue of origin (*Hu et al., 2013*; *Mayers et al., 2016*) or other cell-extrinsic differences between PDAC and LUAD, such as differences in non-cancer cells present in the tumor, drive the difference in TIF composition is another important question for future study.

Tumors in different anatomical locations will also contain different stromal cells and different vascularization that could alter the metabolic microenvironment. Comparison of autochthonous and subcutaneous PDAC tumors driven by the same oncogenic driver mutations revealed significant differences in TIF nutrient levels. This suggests that subcutaneous models of PDAC may not fully recapitulate the metabolic microenvironment of this disease, and is in line with observations that subcutaneous models of PDAC fail to recapitulate many aspects of PDAC tumors including marked stromal infiltration and extracellular matrix deposition (*Hwang et al., 2016*). These observations, along with recent findings that subcutaneous PDAC models do not recapitulate whole body metabolic perturbations observed in autochthonous PDAC models and patients (*Danai et al., 2018*), suggest subcutaneous models do not recapitulate all metabolic aspects of the disease.

As we compared transplant to autochthonous PDAC models, we cannot distinguish between differences in TIF composition driven by anatomical location versus differences caused by unique characteristics of transplant and genetically engineered PDAC models. Nevertheless, it is intriguing that autochthonous PDAC tumors have lower levels of arginine, tryptophan and cystine compared to subcutaneous tumors. Stromal myeloid derived cells have been implicated in depleting each of these nutrients from the tumor microenvironment (*Kumar et al., 2016*), and murine PDAC tumors are known to contain a large number of myeloid lineage cells (*Bayne et al., 2012*; *Goedegebuure et al., 2011*; *Stromnes et al., 2014*; *Zhao et al., 2009*). Additionally, subcutaneous PDAC tumors are known to have fewer stromal cells (*Hwang et al., 2016*; *Sousa et al., 2016*). Thus, it is tempting to speculate that differences in TIF composition between autochthonous and subcutaneous PDAC tumors, including differences in arginine, tryptophan and cystine levels, could be driven by differences in stromal cell populations between anatomical sites. Understanding how anatomical location alters the metabolic microenvironment is critical to determine how local metabolic constraints shape the metabolism of tumors growing in different locations, such as primary and metastatic tumors (*Schild et al., 2018*).

Lastly, we hypothesized that tumor genetics would also alter TIF composition, and studied the effect of *Keap1* loss because *Keap1* null cancer cells have dramatic alterations in cell-intrinsic metabolism (*Best et al., 2018*; *DeNicola et al., 2015*; *DeNicola et al., 2011*; *Mitsuishi et al., 2012*; *Romero et al., 2017*; *Sayin et al., 2017*). Surprisingly, we found no substantial differences in TIF composition between tumors formed from otherwise isogenic LUAD cell lines that differed only in *Keap1* loss. These data argue that factors beyond cancer cell-intrinsic metabolism can be dominant in setting nutrient levels in the microenvironment, although future studies are required to determine how other oncogenic alterations alter TIF composition.

## Implications of TIF nutrient levels for cellular metabolism and function in the tumor

It is tempting to interpret depletion or accumulation of metabolites as simply being due to rapid cancer cell consumption or release of metabolites without appropriate replenishment or removal from the circulation, allowing cancer cell-intrinsic changes in metabolism to be read out as differences in TIF nutrient levels. However, wild type and *Keap1* null cancer cells have significant changes in metabolism (*Best et al., 2018*; *DeNicola et al., 2015*; *DeNicola et al., 2011*; *Mitsuishi et al., 2012*; *Romero et al., 2017*; *Sayin et al., 2017*), yet tumors derived from cells of these genotypes do not show significant alterations in levels of nutrients whose metabolism is known to be altered by *Keap1* loss. Thus, differences in metabolite concentrations between TIF and plasma cannot be used to extrapolate consumption and release of metabolites by tumors. Instead, future experiments measuring differences in arterial and venous metabolite levels could shed light into quantitative tumor nutrient consumption and release (*Gullino et al., 1967*; *Kallinowski et al., 1988*; *Sauer et al., 1982*).

Nutrient levels in the tumor microenvironment can have profound impacts on the metabolism, growth and drug sensitivity of tumor resident cancer cells (*Muir et al., 2018*). Our study provides insight into tumor nutrient levels in vivo and suggests specific implications for tumor resident cell metabolism. First, we find PDAC tumor interstitial fluid is depleted of some nutrients relative to plasma. Interestingly, these depleted nutrients are not necessarily metabolites predicted to be depleted by cell culture studies. For example, PDAC cells consume large amounts of glucose and

glutamine in culture (*Son et al., 2013*; *Ying et al., 2012*), and it has been assumed that these nutrients are depleted in the microenvironment (*Commisso et al., 2013*; *Kamphorst et al., 2015*; *Lyssiotis and Kimmelman, 2017*; *Sherman et al., 2017*; *Sousa et al., 2016*). Branched chain amino acids are also proposed to be limiting for proliferation of PDAC cells in tumors, requiring cancer cells to acquire those nutrients from alternative sources such as stromal cells or extracellular protein (*Grankvist et al., 2018*; *Palm et al., 2015*). We find that neither glutamine nor branched-chain amino acids are substantially depleted in PDAC tumors regardless of anatomical site. Further, while glucose is depleted in PDAC TIF relative to circulatory levels, it is still present at millimolar concentration, and glucose deprivation in TIF is not a universal feature of tumors (*Siska et al., 2017*). Thus, nutrients inferred from cell culture studies to be depleted and limiting in the tumor microenvironment may not be always be key microenvironmental drivers altering PDAC metabolism.

We find that the nutrient most strongly depleted from PDAC TIF is arginine. Arginine supports many aspects of cell physiology (*Morris, 2007*), raising the question of how PDAC cells adapt to survive and proliferate when levels of this amino acid are so low. Many cell types can synthesize arginine using urea cycle enzymes (*Wu et al., 2009*), but these metabolic enzymes are silenced by many tumors to enhance nucleotide production (*Rabinovich et al., 2015*). However, PDAC tumors do not exhibit urea cycle enzyme silencing (*Lee et al., 2018*; *Uhlén et al., 2015*) and are reported to have a functional urea cycle (*Zaytouni et al., 2017*). Perhaps, PDAC cells retain this metabolic pathway to adapt to a tumor microenvironment with limited arginine availability. Additionally, PDAC cells utilize macropinocytosis of environmental protein (*Commisso et al., 2013*; *Kamphorst et al., 2015*), and this route of nutrient acquisition may allow PDAC cells to acquire sufficient arginine to support survival and growth.

TIF composition can alter the function of non-cancer cells, such as immune cells, in the tumor microenvironment (*Buck et al., 2017*). For example, in many tumor types, T lymphocytes can infiltrate and inhibit the progression of tumors, especially when this process is primed with checkpoint blockade treatment (*Ribas and Wolchok, 2018*). In contrast, PDAC tumors are highly immunosuppressive (*Martinez-Bosch et al., 2018*) and immunotherapy shows limited efficacy in this disease (*Hilmi et al., 2018*). Nutrient deprivation in PDAC TIF could contribute to PDAC immunosuppression. Both arginine and tryptophan, which are depleted in the autochthonous PDAC environment, are required for T cell function (*Geiger et al., 2016*; *Moffett and Namboodiri, 2003*). Intriguingly, depletion of myeloid cells capable of degrading arginine and tryptophan from PDAC tumors resulted in increased T cell infiltration, proliferation, and activation in PDAC tumors (*Bayne et al., 2012*; *Stromnes et al., 2014*; *Zhang et al., 2017*). Future studies determining how T cell metabolism and function is impacted by tumor nutrient levels could yield insight into how tumors suppress immune rejection (*Ecker and Riley, 2018*).

Cellular metabolism can respond and adapt to environmental nutrient levels. Indeed, growing cancer cells in media with different nutrient compositions alters their metabolic requirements and response to drugs (*Cantor et al., 2017*; *Muir et al., 2017*; *Palm et al., 2015*; *Schug et al., 2015*). That metabolism is responsive to environment may underpin the limited ability of ex vivo culture models using non-physiological nutrient levels to identify tumor-essential metabolic genes (*Horvath et al., 2016*; *Muir and Vander Heiden, 2018*; *Ryan et al., 2018*). However, given the dearth of information on physiological nutrient levels in solid tumors, it has not been possible to determine the metabolic phenotypes and liabilities of cancer cells in tumor nutrient conditions. By characterizing the polar small molecule nutrients in PDAC and LUAD tumors, media that better approximates the nutrients available to cancer cells in tumors can be formulated. Examining nonpolar and lipid metabolites in the microenvironment will further improve these efforts and lead to cancer models that may better allow us to identify metabolic liabilities of cancer cells that ultimately translate into more effective therapies.

## Materials and methods

### Animal studies

All experiments performed in this study were approved by the MIT Committee on Animal Care (IACUC). All mice in this study were fully backcrossed to the C57BL/6J background. Animals were housed on a 12 hr light and 12 hr dark cycle, with ad libitum access to food and water.

For studies using *Kras*$^{G12D}$ *Trp53*$^{fl/fl}$ *Pdx-1-cre* (KP-/-C) mice (*Bardeesy et al., 2006*), male and female animals of this genotype were allowed for form end-stage tumors, which occurred approximately 8–10 weeks after birth (*Danai et al., 2018*). Animals were then euthanized and tumors harvested for TIF isolation as described below. Tumors weighed between 0.31 g – 2.81 g upon harvesting.

For subcutaneous xenograft studies, 12 week old C57BL/6J animals purchased from Jackson Laboratories (IMSR Cat# JAX:000664, RRID:IMSR_JAX:000664) were injected with 100,000 murine PDAC or LUAD cancer cells (suspended in a volume of 100 μL of Matrigel (Corning, 354234) brought to 10 mg/ml with RPMI-1640 (Corning, 50–020-PC) into the subcutaneous space on the flank of the mice. Cell lines used for subcutaneous engraftment in this study are described below. Tumors were then allowed to grow until they reached ~1 cm$^3$ in volume, which took ~4 weeks after engraftment. Upon the tumor reaching ~1 cm$^3$, animals were euthanized and tumors harvested for TIF isolation as described below.

For dietary studies, C57BL/6J mice were engrafted with murine PDAC cells as described above. On the day of injection, the animals were separated into two cohorts. One group was fed standard mouse chow and the other group was fed a defined amino acid diet (see *Supplementary file 2* for composition of diets). Both groups were fed each diet ad libitum throughout the duration of the experiment. Upon the tumor reaching 1 cm$^3$, animals were euthanized and tumors harvested for TIF isolation as described below.

## Cell lines and culture

The murine PDAC cancer cell line (AL1376) used for making subcutaneous grafts was generated as previously described (*Mayers et al., 2014*) from PDAC tumors from a KP-/-C animal in the C57BL/6J background. The C57BL/6J LUAD cancer cell lines with and without loss of *Keap1* used in tumor grafts in this study were generated previously (*Romero et al., 2017*) from the *Kras*$^{G12D}$ *Trp53*$^{fl/fl}$ *Adenoviral-cre* model of LUAD (*Jackson et al., 2005*; *Jackson et al., 2001*). All cell lines were regularly tested for mycoplasma contamination using the Mycoprobe mycoplasma detection kit (R and D Systems). All cells were cultured in a Heracell (Thermofisher) humidified incubators at 37°C and 5% CO$_2$. Cell lines were routinely maintained in RPMI-1640 (Corning, 50–020-PC) supplemented with 10% heat inactivated fetal bovine serum (Seradigm, Lot 120B14).

## Isolation of tumor interstitial fluid (TIF)

TIF was isolated from tumors using a previously described centrifugal method (*Eil et al., 2016*; *Haslene-Hox et al., 2011*; *Ho et al., 2015*; *Wiig et al., 2003*). Briefly, tumor bearing animals were euthanized by cervical dislocation and tumors were rapidly dissected from the animals. Dissections took <1 min. to complete. Blood was collected from the same animal via cardiac puncture, and was immediately placed in EDTA-tubes (Sarstedt, North Rhine-Westphalia, Germany) and centrifuged at 845 x g for 10 min at 4°C to separate plasma. Plasma was frozen in liquid nitrogen and stored at −80°C until further analysis. Tumors were then weighed and briefly rinsed in room temperature saline (150 mM NaCl) and blotted on filter paper (VWR, Radnor, PA, 28298–020). The entire process of preparing the tumor prior to isolation of TIF took ~2 min. The tumors were then put onto 20 μm nylon filters (Spectrum Labs, Waltham, MA, 148134) affixed atop 50 mL conical tubes, and centrifuged for 10 min. at 4°C at 106 x g. TIF was then collected from the conical tube, frozen in liquid nitrogen and stored at −80°C until further analysis. 5–180 μL of TIF was able to be isolated from ~75% (13/17 tumors) of isolated KP-/-C PDAC tumors. The remaining 25% of tumors did not yield any fluid. The amount of fluid collected per tumor weight is consistent with previous reports of TIF isolation from human tumor samples (*Haslene-Hox et al., 2011*).

## Quantification of lactate dehydrogenase activity in TIF, plasma and tumors

To quantitate the amount of LDH activity present in TIF, plasma, and tumors, we utilized the inherent absorbance of NADH at 340 nm to monitor the generation of lactate from pyruvate and NADH. To an assay buffer composed of 50 mM Tris base (Sigma Aldrich, 93362), 100 mM dithiothreitol (Sigma Aldrich, 646563), 180 μM NADH (Sigma Aldrich, N8129), and 500 μM pyruvate (Sigma Aldrich, P5280), we added 20 μL of sample, then monitored the disappearance over time of absorbance of

light at 340 nM due to the consumption of NADH. A standard curve of LDH (Sigma-Aldrich, 10127230001, E.C. 1.1.1.27) was generated with points of 0, 0.005838, 0.007783, 0.011675, 0.02335, and 0.0467 units of LDH activity by diluting LDH in Tris buffered saline composed of 50 mM Tris HCl (VWR, 4103) and 150 mM NaCl (Sigma Aldrich, 746398) pH adjusted to 7.5. Whole tumor samples were prepared by homogenization using a mortar and pestle submerged in liquid nitrogen, and the resulting powder was resuspended at 10 mg/mL in Tris buffered saline. Plasma samples were added undiluted, and TIF samples were diluted 1:10 in Tris buffered saline. Based on the slope of the LDH standard curve, the amount of LDH activity in each sample was calculated and corrected for dilution.

## Quantification of metabolite levels in TIF and plasma

In order to quantitate metabolites in TIF and plasma samples, we first constructed a library of 149 chemical standards of plasma polar metabolites (see *Supplementary file 1* for suppliers for each chemical standard). These compounds were selected to encompass a number of metabolic processes and have previously been included in efforts to profile plasma polar metabolites by LC/MS (*Cantor et al., 2017*; *Evans et al., 2009*; *Lawton et al., 2008*; *Mazzone et al., 2016*). We pooled these metabolites into seven separate chemical standard pools (*Supplementary file 1*). To do this, each metabolite in a given pool was weighed and then mixed (6 cycles of 1 min. mixing at 25 Hz followed by 3 min. resting) using a Mixer Mill MM301 (Retsch, Düsseldorf, Germany), and mixed metabolite powder stocks were stored at −20°C prior to resuspension and analysis. Stock solutions of the mixed standards pools containing ~5 mM,~1 mM,~300 µM,~100 µM,~30 µM,~10 µM,~3 µM and ~1 µM of each metabolite were made in HPLC grade water and were stored at −80°C (see *Supplementary file 1* for the concentration of each metabolite in the external standard pools). We refer to these stock solutions as 'external standard pools' throughout. External standard pools were used to confirm the retention time and *m/z* for each analyte and provide standards to quantitate concentrations of stable isotope labeled internal standards used in downstream analysis, as well as to quantitate metabolite concentrations in TIF and plasma samples directly where internal standards were not available (see below for details).

We had three classes of samples: plasma, TIF and the external standard pool dilutions (prepared in water). We extracted polar metabolites (*Figure 1—figure supplement 2*) from each sample type using the same extraction mix and protocol: 5 µL of sample (plasma, TIF or external standard pool dilution) was mixed with 45 uL of acetonitrile:methanol:formic acid (75:25:0.1) extraction mix including the following isotopically labeled internal standards: $^{13}C$ labeled yeast extract (Cambridge Isotope Laboratory, Andover, MA, ISO1), $^{13}C_3$ lactate (Sigma Aldrich, Darmstadt, Germany, 485926), $^{13}C_3$ glycerol (Cambridge Isotope Laboratory, Andover, MA, CLM-1510), $^{13}C_6$ $^{15}N_2$ cystine (Cambridge Isotope Laboratory, Andover, MA, CNLM-4244), $^2H_9$ choline (Cambridge Isotope Laboratory, Andover, MA, DLM-549), $^{13}C_4$ 3-hydroxybutyrate (Cambridge Isotope Laboratory, Andover, MA, CLM-3853), $^{13}C_6$ glucose (Cambridge Isotope Laboratory, Andover, MA, CLM-1396), $^{13}C_2$ $^{15}N$ taurine (Cambridge Isotope Laboratory, Andover, MA, CNLM-10253), $^2H_3$ creatinine (Cambridge Isotope Laboratory, Andover, MA, DLM-3653), 8-$^{13}C$ adenine (Cambridge Isotope Laboratory, Andover, MA, CLM-1654), $^{13}C_5$ hypoxanthine (Cambridge Isotope Laboratory, Andover, MA, CLM-8042), 8-$^{13}C$ guanine (Cambridge Isotope Laboratory, Andover, MA, CLM-1019), $^{13}C_3$ serine (Cambridge Isotope Laboratory, Andover, MA, CLM-1574) and $^{13}C_2$ glycine (Cambridge Isotope Laboratory, Andover, MA, CLM-1017). All solvents used in the extraction mix were HPLC grade. Samples were then vortexed for 10 min. at 4°C and insoluble material was sedimented by centrifugation at 15 kg for 10 min. at 4°C. 20 µL of the soluble polar metabolite extract was taken for LC/MS analysis.

LC/MS analysis was performed on the sample extracts using a QExactive orbitrap mass spectrometer using an Ion Max source and heated electrospray ionization (HESI) probe coupled to a Dionex Ultimate 3000 UPLC system (Thermo Fisher Scientific, Waltham, MA). External mass calibration was performed every 7 days, and internal mass calibration (lock masses) was not used. 2 µL of each sample was injected onto a ZIC-pHILIC 2.1 × 150 mm analytical column equipped with a 2.1 × 20 mm guard column (both 5 µm particle size, EMD Millipore). The autosampler and column oven were held at 4°C and 25°C, respectively. Buffer A was 20 mM ammonium carbonate, 0.1% ammonium hydroxide; buffer B was acetonitrile. The chromatographic gradient was run at a flow rate of 0.150 mL/min as follows: 0–20 min: linear gradient from 80% to 20% B; 20–20.5 min: linear gradient from 20% to 80% B; 20.5–28 min: hold at 80% B. The mass spectrometer was operated in full scan, polarity-

switching mode with the spray voltage set to 3.0 kV, the heated capillary held at 275°C, and the HESI probe held at 350°C. The sheath gas flow rate was set to 40 units, the auxiliary gas flow was set to 15 units, and the sweep gas flow was set to one unit. The MS data acquisition was performed in a range of 70–1000 m/z, with the resolution set to 70,000, the AGC target at 1e6, and the maximum injection time at 20 msec.

After LC/MS analysis, metabolite identification was performed with XCalibur 2.2 software (Thermo Fisher Scientific, Waltham, MA) using a 5ppm mass accuracy and a 0.5 min. retention time window. For metabolite identification, external standard pools were used for assignment of metabolites to peaks at given m/z and retention time, and to determine the limit of detection for each metabolite, which ranged from 100 nM to 3 μM (see *Supplementary file 1* for the m/z, retention time and limit of detection for each metabolite analyzed).

After metabolite identification, quantification was performed by two separate methods for either quantification by stable isotope dilution or external standard calibration (*Figure 1—figure supplement 3*). For quantification by stable isotope dilution, where internal standards were available, we first compared the peak areas of the stable isotope labeled internal standards with the external standard pools diluted at known concentrations. This allowed for quantification of the concentration of labeled internal standards in the extraction mix. Subsequently, we compared the peak area of a given unlabeled metabolite in the TIF and plasma samples with the peak area of the now quantified internal standard to determine the concentration of that metabolite in the TIF or plasma sample. 70 metabolites were quantitated using this internal standard method (see *Supplementary file 1* for the metabolites quantitated with internal standards).

For metabolites without internal standards, quantification by external calibration was performed as described below (*Figure 1—figure supplement 3*). First, the peak area of each externally calibrated analyte was normalized to the peak area of a labeled amino acid internal standard that eluted at roughly the same retention time to account for differences in recovery between samples (see *Supplementary file 1* for the labeled amino acid paired to each metabolite analyzed without an internal standard). This normalization was performed in both biological samples and external standard pool dilutions. From the normalized peak areas of metabolites in the external standard pool dilutions, we generated a standard curve describing the relationship between metabolite concentration and normalized peak area. The standard curves were linear with fits typically at or above $r^2 = 0.95$. Metabolites which did not meet these criteria were excluded from further analysis. These equations were then used to convert normalized peak areas of analytes in the TIF or plasma samples into analyte concentration in the samples. 74 metabolites were quantitated using this method. The relationship between metabolite concentration and normalized peak area is matrix dependent, and the external standards are prepared in water, which is a different matrix than either TIF or plasma. Therefore, we consider metabolite measurements using this external standard method semi-quantitative.

## Measurement of relative intratumoral metabolite levels

KP-/-C tumors were removed from animals as for TIF isolation. These tumors were then cut in half. One half was immediately cryogenically frozen using a BioSqueezer (BioSpec, Bartlesville, OK) cooled with liquid nitrogen. The other was subjected to TIF isolation as described above, and then similarly cryogenically frozen. Frozen tumor pieces were then then ground to a fine homogenous powder using a mortar and pestle cooled with liquid nitrogen. 15–32 mg of tumor powder was weighed into sample tubes, and metabolites were then extracted using 600 μL HPLC grade methanol (Sigma-Aldrich), 300 μL HPLC grade water (Sigma-Aldrich), and 400 μL chloroform (Sigma-Aldrich). The extraction mix included the following internal standards: 200 μM $^{13}C$ and $^{15}N$ labeled amino acids (Cambridge Isotope Laboratory, Andover, MA, MSK-A2-1.2), 200 μM $^{13}C_4$ succinate (Cambridge Isotope Laboratory, Andover, MA, CLM-1571-PK) and 200 μM $^{13}C_5$ hypoxanthine (Cambridge Isotope Laboratory, Andover, MA, CLM-8042). Samples were vortexed for 10 min at 4°C, then centrifuged at 21000 x g at 4°C for 10 min. 400 μL of the aqueous layer was removed and dried under nitrogen.

Dried tumor extracts were resuspended in 100 μL HPLC grade water and LC/MS was performed as described above for TIF and plasma samples. Relative quantitation of 112 detected metabolites (see *Figure 1—figure supplement 1—source data 1* for detected metabolites) was performed with XCalibur QuanBrowser 2.2 (Thermo Fisher Scientific, Waltham, MA) using a five ppm mass tolerance

and referencing an in-house library of chemical standards for identification of metabolites by exact mass and retention time. Hypoxanthine and xanthine peak areas were normalized to the $^{13}C_5$ hypoxanthine internal standard and succinate was normalized to the $^{13}C_4$ succinate peak area. All other metabolites were normalized to the $^{13}C_5$ $^{15}N_1$ leucine internal standard. All metabolites were normalized to the mass of tumor powder from which metabolites were extracted.

### Statistical analysis of TIF and plasma metabolite levels

After determining the concentration of each metabolite in each plasma or TIF sample, all multivariate statistical analysis on the data was performed using Metaboanalyst 4.0 (*Chong et al., 2018*). All metabolite concentrations were not normalized prior to analysis, but the data was auto-scaled (mean-centered and divided by the standard deviation of each variable) prior to analysis, as this method of scaling has been shown to perform well with metabolomics data (*van den Berg et al., 2006*). After scaling the data, we performed principal component analysis and hierarchical clustering with Euclidean distance measurement and clustering by the Ward algorithm. Univariate analysis was performed comparing metabolite levels between groups where metabolite differences of interest were defined by a fold change greater than 1.5 and significance as a FDR-adjusted P-value less than 0.1 assuming unequal group variance. All other statistical analysis and graphical representation of data was performed as described in the Results using GraphPad Prism 7 (GraphPad Software, La Jolla, CA).

## Acknowledgements

We thank Aaron Hosios and all members of the Vander Heiden lab for many useful discussions and experimental advice, and Allison Lau and Sharanya Sivanand for supplying tumor bearing animals used in this work. We thank Sarah Price for assistance with developing data analysis tools. We thank Peggy Hsu, Alicia Darnell, and Allison Lau for comments and help with editing the manuscript. We thank Rodrigo Romero and Tyler Jacks for generously supplying murine LUAD cell lines and providing experimental advice. We thank Susan Kaech, Victor Chubakov and Thomas Roddy for helpful discussions on interstitial fluid isolation and analysis. Lastly, we thank Cambridge Isotope Laboratories for the generous gift of isotopically labeled yeast extract used in this work. This work was supported by grants to MGVH from the NIH (R01CA168653, R01CA201276, and P30CA1405141), the Lustgarten Foundation, the MIT Center for Precision Cancer Medicine, SU2C, and the Ludwig Center at MIT. AM and LVD were supported by NIH Ruth Kirschstein Fellowships, F32CA213810 and F32CA210421 respectively. MRS was supported by T32GM007287 and acknowledges additional support from an MIT Koch Institute Graduate Fellowship. DYG received support from T32GM007753. MGVH is a Howard Hughes Medical Institute Faculty Scholar.

## Additional information

### Competing interests

Matthew G Vander Heiden: Reviewing editor, *eLife;* has applied for patents for therapeutic strategies to target cancer metabolism (US Patent App. 15/890,220); Advisor for Agios Pharmaceuticals, Aeglea Biotherapeutics, Auron Therapeutics. Laura V Danai, Dan Y Gui, Alexander Muir: has applied for patents for therapeutic strategies to target cancer metabolism (US Patent App. 15/890,220). The other authors declare that no competing interests exist.

### Funding

| Funder | Grant reference number | Author |
|---|---|---|
| Koch Institute for Integrative Cancer Research | Koch Institute Graduate Fellowship | Mark R Sullivan<br>Laura V Danai<br>Dan Y Gui<br>Matthew G Vander Heiden<br>Alexander Muir |

| | | |
|---|---|---|
| National Cancer Institute | F32CA210421 | Mark R Sullivan<br>Laura V Danai<br>Dan Y Gui<br>Matthew G Vander Heiden<br>Alexander Muir |
| Ludwig Institute for Cancer Research | | Mark R Sullivan |
| Lustgarten Foundation | | Laura V Danai<br>Alexander Muir |
| Howard Hughes Medical Institute | | Matthew G Vander Heiden |
| Stand Up To Cancer | | Matthew G Vander Heiden |
| National Cancer Institute | R01CA168653 | Matthew G Vander Heiden |
| National Cancer Institute | R01CA201276 | Matthew G Vander Heiden |
| National Cancer Institute | P30CA1405141 | Matthew G Vander Heiden |
| National Cancer Institute | F32CA213810 | Alexander Muir |

The funders had no role in study design, data collection and interpretation, or the decision to submit the work for publication.

## Author contributions

Mark R Sullivan, Alexander Muir, Conceptualization, Formal analysis, Funding acquisition, Investigation, Visualization, Methodology, Writing—original draft, Writing—review and editing; Laura V Danai, Funding acquisition, Investigation, Methodology, Writing—review and editing; Caroline A Lewis, Formal analysis, Supervision, Investigation, Methodology, Writing—review and editing; Sze Ham Chan, Formal analysis, Investigation, Methodology; Dan Y Gui, Conceptualization, Investigation, Writing—review and editing; Tenzin Kunchok, Emily A Dennstedt, Investigation, Methodology; Matthew G Vander Heiden, Conceptualization, Supervision, Funding acquisition, Writing—review and editing

## Author ORCIDs

Mark R Sullivan ⓘ http://orcid.org/0000-0001-5765-0459
Caroline A Lewis ⓘ http://orcid.org/0000-0003-1787-5084
Matthew G Vander Heiden ⓘ https://orcid.org/0000-0002-6702-4192
Alexander Muir ⓘ https://orcid.org/0000-0003-3811-3054

## Ethics

Animal experimentation: This study was performed in accordance with the recommendations in the Guide for the Care and Use of Laboratory Animals. All animals experiments were performed using protocols (#1115-110-18) that were approved by the MIT Committee on Animal Care (IACUC). All surgeries were performed using isoflurane anesthesia administered by vaporizer and every effort was made to minimize suffering.

## Decision letter and Author response

Decision letter https://doi.org/10.7554/eLife.44235.046
Author response https://doi.org/10.7554/eLife.44235.047

# Additional files

## Supplementary files

• Supplementary file 1. Information on metabolites analyzed in this study including metabolite name, organization of metabolite pools, concentration of each metabolite in each metabolite pool, polarity in which the metabolite was best detected, *m/z*, limit of detection, and method of quantification.
DOI: https://doi.org/10.7554/eLife.44235.040

• Supplementary file 2. Composition of animal diets used in this study.
DOI: https://doi.org/10.7554/eLife.44235.041

• Transparent reporting form
DOI: https://doi.org/10.7554/eLife.44235.042

## Data availability

Source data files detailing the concentrations of each metabolite in each sample are included for all figures. We have also deposited this information and the raw mass spectra in Metabolomics Workbench (http://www.metabolomicsworkbench.org/) as project ID: PR000750.

The following dataset was generated:

| Author(s) | Year | Dataset title | Dataset URL | Database and Identifier |
|---|---|---|---|---|
| Muir A | 2019 | Quantification of microenvironmental metabolites in murine cancers reveals determinants of tumor nutrient availability | http://doi.org/10.21228/M8QX2G | Metabolomics Workbench, 10.21228/M8QX2G |

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
