## [Decision Letter]

Thank you for submitting your article "Quantification of microenvironmental metabolites in murine cancers reveals determinants of tumor nutrient availability" for consideration by *eLife*. Your article has been reviewed by three peer reviewers, including Ralph DeBerardinis as the Reviewing Editor and Reviewer #1, and the evaluation has been overseen by Sean Morrison as the Senior Editor. The following individual involved in review of your submission has agreed to reveal their identity: Christian Frezza (Reviewer #2).

The reviewers have discussed the reviews with one another and the Reviewing Editor has drafted this decision to help you prepare a revised submission. Also, please note that while all reviewers saw value in Essential revisions 6 and 7, we appreciate that it may take more than 2 months to perform these experiments. We encourage you to include any such data you may have acquired or could acquire within 2 months to address these two points, but we would likely proceed with the paper even without these additional models, as long as the other points can be thoroughly addressed.

Summary:

Sullivan et al. applied quantitative metabolomics to measure metabolites in tumor interstitial fluid (TIF) from mice. They show that TIF metabolite content differs from plasma and is influenced by anatomic location, diet and, to a lesser extent, genetic factors. Altogether, the authors provide evidence that both tumor-intrinsic and tumor-extrinsic properties influence the metabolic makeup of the TIF, and they present a quantitative method to study this problem. The work is timely and the intriguing findings will present a valuable resource for the field of cancer metabolism.

Essential revisions:

1) Although the authors were careful in isolating TIF, questions about artifacts will inevitably arise. Some additional discussion and (ideally) controls around these issues would be helpful. First, the authors carefully control against contamination of intracellular fluid in their prep, but it is unclear how or whether they know if the samples experience exchanges of metabolites into and out of TIF during the prep, thereby changing the apparent TIF concentrations. Some discussion around this issue would increase confidence in the findings. Second, can the authors clarify whether the procedure, which involves a period of ischemia, affects the tumor metabolome? For example, can the authors compare the metabolic signature of a freshly isolated tumor to one after the isolation of the TIF? Measurement of succinate and xanthine/hypoxanthine would help indicate the degree of ischemia experienced by the tissue.

2) The authors state that metabolite levels in normal tissues more closely match circulating plasma concentrations. They should measure interstitial metabolites in normal pancreas and other relevant tissues and compare them to the changes the observe in TIF.

3) Given the setup in the Introduction about high rates of glucose fermentation to lactate by tumors, it is relevant that lactate levels were not increased in the TIF in most experiments. I think readers would be interested in this finding, because it impacts how we think about glucose handling in these tumors. Could the authors show the lactate data in Figure 2?

4) Most plasma vs TIF comparisons have a very different number of replicates per group. It may not have been possible to collect TIF from all tumors, but it would be helpful to illustrate the metabolic differences between matched plasma and TIF in the same mouse, perhaps as a supplementary figure.

5) A related issue is the statistical tools used to compare TIF and plasma, because the unbalanced number of replicates may lead to erroneous p-values if not properly accounted for. It is not clear from the Materials and methods which statistical test was applied for these comparison (e.g. Figure 2D-J) and how the assumptions of the test were confirmed (e.g. normality of distribution, homogeneity of variance, etc.).

6) The variation between subcutaneous and autochthonous tumor metabolites is intriguing. The manuscript and the field would benefit if orthotopic allografts were also included in this comparison.

7) The authors speculate that cell intrinsic metabolic states and extrinsic factors such as circulation restriction by the stroma or high interstitial pressure exert major effects on TIF content. Some of these variables could be experimentally tested. For example, stroma can be depleted in KPC tumors, and interstitial pressure could be manipulated pharmacologically. These data, if available, would improve the paper and benefit the field.

---

## [Author Response]

Essential revisions:1) Although the authors were careful in isolating TIF, questions about artifacts will inevitably arise. Some additional discussion and (ideally) controls around these issues would be helpful. First, the authors carefully control against contamination of intracellular fluid in their prep, but it is unclear how or whether they know if the samples experience exchanges of metabolites into and out of TIF during the prep, thereby changing the apparent TIF concentrations. Some discussion around this issue would increase confidence in the findings. Second, can the authors clarify whether the procedure, which involves a period of ischemia, affects the tumor metabolome? For example, can the authors compare the metabolic signature of a freshly isolated tumor to one after the isolation of the TIF? Measurement of succinate and xanthine/hypoxanthine would help indicate the degree of ischemia experienced by the tissue.

We agree that while we have controlled for TIF contamination by intracellular material during TIF isolation, we had not addressed changes in TIF metabolite composition that could occur during the time that elapses during TIF harvest. It is difficult to perfectly control for how time required to harvest TIF affects TIF metabolite composition, because we do not currently have a method that allows for faster TIF isolation. We appreciate that a faster isolation method would allow us to make such a comparison and determine if any changes in TIF composition occur during the time elapsed when TIF is isolated by centrifugation. We are actively developing methods (e.g. TIF isolation using Guyton capsules PMID: 14190544) that would allow instantaneous TIF isolation, but developing these techniques requires extensive engineering and animal experiments that are not possible in a short time frame. Nevertheless, we agree this is an important point and at the reviewers’ suggestion, we have added additional discussion on the potential limitations of TIF isolation by centrifugation in the Results section where we introduce the method.

To address the reviewers’ second point, we have performed an additional experiment to determine the extent to which TIF isolation by centrifugation results in tumor ischemia and alterations in intratumoral metabolite levels, which may also influence what we measure in TIF. For this study, we isolated KP-/-C PDAC tumors, and cut the tumors in half. One half was immediately flash frozen, while TIF was isolated from the other half. The TIF-isolated half was then flash frozen. We then extracted metabolites from both tumor samples and compared relative metabolite levels (including the ischemia markers succinate, xanthine and hypoxanthine) between the matched fresh and TIF-isolated PDAC tumors. Fresh and TIF-isolate tumor samples did not cluster separately by principal component analysis of relative metabolite levels. We did observe small increases in xanthine and hypoxanthine levels in TIF-isolated tumor halves, but no increase in succinate levels. The increases in xanthine and hypoxanthine were much lower than observed previously in ischemic tissues (PMID: 25383517). These data suggest that the tumors likely experience a small degree of ischemia during the time needed to isolate TIF, but that this does not lead to significant changes in most intratumoral metabolite levels. While it is impossible to know how the TIF isolation employed affects the composition of the TIF itself, given the small degree of accumulation of ischemic markers in the tumors and that analysis of multiple metabolite levels between TIF-isolated and fresh tumors does not show stark differences, we suspect it is unlikely that ischemia experienced by the tumor during centrifugation substantially alters TIF composition. Consideration of these issues is also discussed in the revised manuscript, with the data included as Figure 1—figure supplement 1 and the raw relative abundances of metabolites included in Figure 1—source data 3.

2) The authors state that metabolite levels in normal tissues more closely match circulating plasma concentrations. They should measure interstitial metabolites in normal pancreas and other relevant tissues and compare them to the changes the observe in TIF.

We agree with the reviewers that measuring the interstitial fluid of normal tissues and comparing this to tumor interstitial fluid would be very informative. We have repeatedly attempted to isolate interstitial fluid from normal murine organs using the same centrifugation method that we used to isolate TIF. Unfortunately, we were unable to isolate interstitial fluid from the normal organs. This could be due to a number of factors including the small size of mouse organs, and the lower interstitial volume and pressure of normal organs (PMID: 3555767). Thus, comparing TIF to normal tissue IF will require additional method development, including development and validation of another isolation method, that will take a substantial effort to complete. We nevertheless appreciate the point, and to address the reviewers’ comment we have added additional discussion to the Results section that we were unable isolate murine tissue interstitial fluid, limiting the comparisons we can make at this time to TIF and the bulk circulation.

3) Given the setup in the Introduction about high rates of glucose fermentation to lactate by tumors, it is relevant that lactate levels were not increased in the TIF in most experiments. I think readers would be interested in this finding, because it impacts how we think about glucose handling in these tumors. Could the authors show the lactate data in Figure 2?

We agree with the reviewers that this is an interesting observation, and we have included the lactate data in Figure 2.

4) Most plasma vs TIF comparisons have a very different number of replicates per group. It may not have been possible to collect TIF from all tumors, but it would be helpful to illustrate the metabolic differences between matched plasma and TIF in the same mouse, perhaps as a supplementary figure.

We agree that analysis of matched plasma and TIF samples is informative for interpreting the data. We have now included the same analysis as performed in Figure 2 using only matched PDAC TIF and plasma samples. The results are shown in Figure 2—figure supplement 2. We note that the same differences between plasma and TIF we observed between all PDAC TIF and plasma samples are largely mirrored in analysis of only the paired samples.

5) A related issue is the statistical tools used to compare TIF and plasma, because the unbalanced number of replicates may lead to erroneous p-values if not properly accounted for. It is not clear from the Materials and methods which statistical test was applied for these comparison (e.g. Figure 2D-J) and how the assumptions of the test were confirmed (e.g. normality of distribution, homogeneity of variance, etc.).

We thank the reviewers for identifying this concern and apologize that it was not sufficiently clear which statistical analyses we performed for comparisons of individual metabolite levels. The unpaired two-tailed Welch’s t-test we were using in Figure 2, Figure 3 and Figure 6 accounts for unequal variance and sample number between groups, but does rely upon normality of distribution, which we did not test. To address this issue, we used the Shapiro-Wilk test to test for normality of distribution for all individual metabolite comparisons shown throughout. The distributions were largely normal, and where this was the case we used the same unpaired two-tailed Welch’s t-test now knowing that the condition of normality of distribution was met. Where the distribution was not normal, we used Wilcoxon matched-pairs signed rank test to test for significance, as this test does not require normality of distribution. We have updated the figure legends to indicate exactly which tests were utilized for each metabolite comparisons presented. Importantly, applying the correct statistical tests did not change which differences were statistically insignificant and did not alter our interpretation of the data.

6) The variation between subcutaneous and autochthonous tumor metabolites is intriguing. The manuscript and the field would benefit if orthotopic allografts were also included in this comparison.

We agree that comparison of TIF from different PDAC models, including orthotopic allografts, would be useful in helping researchers select appropriate mouse models for their studies. We also agree that it would strengthen the conclusion that we have drawn from the data that anatomical location could impact the tumor nutrient environment. Unfortunately, generating and isolating TIF from PDAC orthotopic allograft tumors was not possible in the time allowed for revision, but are working on addressing this question to better understand how anatomical location influences TIF composition, and to determine whether allograft models recapitulate the TIF microenvironment of autochthonous tumors. We have also included discussion of this point in the Results section of the revised manuscript.

7) The authors speculate that cell intrinsic metabolic states and extrinsic factors such as circulation restriction by the stroma or high interstitial pressure exert major effects on TIF content. Some of these variables could be experimentally tested. For example, stroma can be depleted in KPC tumors, and interstitial pressure could be manipulated pharmacologically. These data, if available, would improve the paper and benefit the field.

We agree with the reviewers that some cancer cell intrinsic and extrinsic factors that we speculate alter TIF composition could be experimentally tested. We anticipate studies that manipulate tumor cells, stromal cells and vasculature will shed more light on what factors influence the metabolic composition of tumors. As noted in the decision letter, these experiments are complicated and require commitment of time and effort that was not possible within the timeframe of the revision. However, we do think this is a critical point that warrants future studies, and have thus included a new section on this point in the Discussion of the revised manuscript.